# NegMerge: Consensual Weight Negation for Strong Machine Unlearning

## Abstract

Machine unlearning aims to selectively remove specific knowledge from a model. Current methods, such as task arithmetic, rely on fine-tuning models on the forget set, generating a task vector, and subtracting it from the original model. However, we argue the effectiveness of this approach is highly sensitive to hyperparameter selection, necessitating careful validation to identify the best model among many fine-tuned candidates. In this paper, we propose a novel method that leverages all given fine-tuned models rather than selecting a single one. By constructing task vectors from models trained with varied hyperparameters and merging only the components of the task vectors with consistent signs, we perform unlearning by negating the merged task vector from the original model. Given that existing methods also utilize multiple fine-tuned models, our approach delivers more effective unlearning without incurring additional computational costs. We demonstrate the effectiveness of our method on both vision-language models and standard image classification models, showing improved unlearning performance with minimal degradation on the retain set, outperforming state-of-the-art techniques.

## 1 Introduction

Recent advances in pre-training (Devlin, 2018; Dosovitskiy et al., 2021; Radford et al., 2021; Oquab et al., 2023; Achiam et al., 2023; Liu et al., 2024) have achieved remarkable performance, primarily driven by the use of large-scale datasets. However, the datasets often include underfiltered, unwanted, or sensitive private information, which raises critical concerns about privacy protection. The *Right to be Forgotten* regulation (Hoofnagle et al., 2019) allows individuals to request the deletion of their personal data. However, applying this concept to machine learning models is challenging because the training process deeply embeds the data into the model's parameters, making it difficult to remove its influence. The most straightforward solution is to remove the data from the training set and retrain the model from scratch, which requires enormous computational resources. As a result, ensuring that models forget learned patterns becomes a challenging task. *Machine unlearning* (Warnecke et al., 2021; Golatkar et al., 2020; Thudi et al., 2022; Koh & Liang, 2017; Jia et al., 2023; Chen et al., 2023; Fan et al., 2023) offers a solution by enabling models to erase specific knowledge without the need for full retraining.

Despite promising results, many existing methods struggle to remove only the target knowledge while preserving the rest. This challenge arises because fine-tuning often disrupts knowledge in the *retain set* (i.e., remaining data) during attempts to erase knowledge from the *forget set* (i.e., data to be forgotten) (Chen et al., 2023; Fan et al., 2023). A known method robust to this issue is task arithmetic (Ilharco et al., 2022a), where direct fine-tuning of the model is avoided. Instead, this method calculates a task vector – the parameter-wise difference between the original model and a model fine-tuned on the forget set. The task vector is then subtracted from the original model through a negation operation. This process, referred to as *forgetting by negation*, has demonstrated strong unlearning performance while preserving the model's knowledge, similar to continual learning researches (Kirkpatrick et al., 2017; Aljundi et al., 2018) addressing catastrophic forgetting (Kirkpatrick et al., 2017). However, we argue that task arithmetic has limitations; not all fine-tuned models are suitable for task vectors, and thus, unlearning performance is highly sensitive to hyperparameter setups used for fine-tuning. As a result, searching for an optimal hyperparameter set for effective unlearning can be both time-consuming and computationally costly.

To address these limitations, we propose a novel method, NegMerge, that improves the process of forgetting by negation. We argue that relying on a single optimal model, as current methods (Ilharco et al., 2022a; Ortiz-Jimenez et al., 2024) do, is not truly optimal. Hyperparameter tuning generates multiple fine-tuned models, and instead of selecting just one, we suggest leveraging all of them. Specifically, we compute the final task vector by merging multiple task vectors derived from the fine-tuned models. This approach draws inspiration from model merging techniques (Wortsman et al., 2022; Yang et al., 2023; Jang et al., 2024), which similarly utilize multiple fine-tuned models to enhance performance. By extending this concept to machine unlearning, we provide a more effective solution. Specifically, unlike these existing techniques, we only combine elements with consistent signs across the task vectors while masking elements with inconsistent signs to zero.

We demonstrate the effectiveness of our approach in two experimental settings. The first involves unlearning specific knowledge from a vision-language model like CLIP (Radford et al., 2021). The second focuses on unlearning knowledge from specific data points in a general image classification network (Chen et al., 2023; Fan et al., 2023). We validate our method using the ViT (Dosovitskiy et al., 2021) and ResNet (He et al., 2016) architectures across nine datasets. In both settings, our approach achieves new state-of-the-art performance while using similar or fewer computational resources than existing methods.

## 2 RELATED WORK

**Machine Unlearning.** Recent machine unlearning methods can be categorized into two main groups; unlearning specific knowledge in vision-language pre-trained models (Ilharco et al., 2022a; Ortiz-Jimenez et al., 2024) and unlearning data in standard classification networks (Chen et al., 2023; Fan et al., 2023). Traditionally, these categories have been viewed as separate fields.

In the formal setup, the negation method in task arithmetic (Ilharco et al., 2022a) is commonly used for unlearning specific knowledge. A recent advancement is the neural tangent kernel-based linear negation method (Ortiz-Jimenez et al., 2024), which addresses weight disentanglement issues in task arithmetic by linearizing models and fine-tuning them in their tangent space. Both techniques depend on a single fine-tuned model to compute the task vector.

On the other hand, unlearning with a standard image classifier usually involves fine-tuning the original model. Fine-tuning (Warnecke et al., 2021) and $\ell_1$-sparse (Jia et al., 2023) aim to overfit the model only on the retain set to erase the knowledge of the forget set. Meanwhile, Influence (Koh & Liang, 2017) and SalUn (Fan et al., 2023) utilize both the retain and forget sets to selectively degrade performance on the forget set while maintaining it on the retain set.

When the forget set is much smaller than the retain set, using retain set for unlearning can be inefficient. This challenge has led to the development of methods that focus on unlearning using only the forget set. Several approaches (Golatkar et al., 2020; Chen et al., 2023) attempt this by relabeling the forget set to different classes and fine-tuning the model. However, these methods often suffer from catastrophic forgetting of the retain set, as the retain set is not used during fine-tuning.

This paper proposes a unified approach to tackle both classification tasks using vision-language models and standard models. Furthermore, our method also focuses on using only the forget set for unlearning. We recognize the inherent trade-off between unlearning performance and retaining performance on the retain set. Relying on a single model to address this trade-off is inefficient. To overcome this, we propose a new approach that utilizes multiple fine-tuned models. By building on task arithmetic, our method computes a more effective task vector from these models, enhancing unlearning performance.

**Model Merging.** The concept of Model soups (Wortsman et al., 2022) addresses inefficiencies in the validation process, where many models are discarded, and only the best one is retained. This approach advocates for merging the weights of sub-optimal models to enhance generalization performance without additional computational demands. Following this insight, more advanced model merging techniques have emerged. Task Arithmetic (Ilharco et al., 2022a) introduces the task vector, demonstrating that merging these vectors can effectively enhance a model's multi-task capabilities. TIES-Merging (Yadav et al., 2024) refines the merging process by incorporating a trimming step and, in cases of sign conflicts, selects one sign through a voting process, merging all task vectors corresponding to the chosen sign. AdaMerging (Yang et al., 2024) autonomously learns

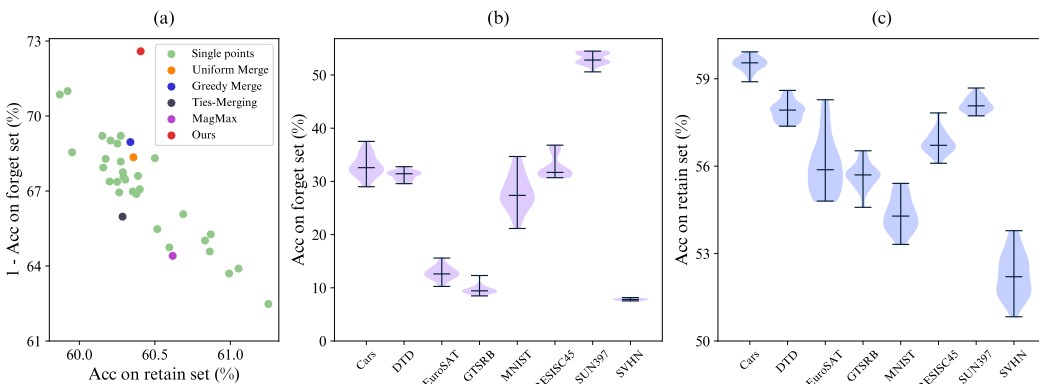

Figure 1: **Hyperparameter Sensitivity in Negation Methods.** We first share our motivation for this work. (a): Each point represents the accuracy of the forget set and the retain set. For the forget set, 1 - accuracy on the forget set is used for better visibility. The green points indicate the results of models fine-tuned with various hyperparameters, while the points in other colors denote results from different methods, including ours. This experiment uses the CLIP ViT-B-32 model on the Cars dataset (Krause et al., 2013). (b), (c): The accuracy distribution for different hyperparameter choices on the forget and retain set, respectively. We observe that 1) the models trained under varied hyperparameters exhibit different unlearning capabilities; 2) smartly utilizing them could improve the capability without concerns on hyperparameter sensitivity (ours).

the coefficients for merging models, either task-wise or layer-wise, and does so without depending on the original training data. MagMax (Marczak et al., 2024) selects task vector elements based on their largest magnitudes.

## 3 METHOD

### 3.1 BACKGROUND

**Task Arithmetic.** Task arithmetic (Ilharco et al., 2022a) defines a *task vector* $\tau_t = \theta_{ft}^t - \theta_{pre}$. Specifically, the vectors are the result of subtracting (negating) the weights of a pre-trained model $\theta_{pre}$ from those of a model $\theta_{ft}^t$ fine-tuned on a target task $t$. We can adjust the model in the desired direction by adding or subtracting the sum of these task vectors $\tau = \sum_t \tau_t$ from the original model's weights, according to the formula $\theta_{new} = \theta_{pre} + \lambda\tau$. This approach is more computationally efficient than fine-tuning, as it leverages pre-trained models from public repositories and eliminates the need for additional training.

A key application of task arithmetic is to make a model forget certain capabilities (Ilharco et al., 2022a). This can be achieved through the *negation* of task vectors from the original weight, which decreases performance on a target task. For instance, task arithmetic can be applied to unlearning in models like CLIP (Radford et al., 2021), which is a strong vision-language model. In the original paper, the authors demonstrated that task vectors derived from a CLIP model fine-tuned on a specific dataset (*e.g.*, Cars) could reduce the model's accuracy on the fine-tuning dataset while maintaining overall accuracy on a general dataset (*e.g.*, ImageNet). However, while task arithmetic has shown promising results for machine unlearning, there has been little research on fine-tuning models and computing task vectors for more effective unlearning. Our research addresses this gap.

**Motivation.** Our pilot study identifies two major challenges. First, unlearning performance is highly sensitive to the hyperparameters used for fine-tuning. Figures 1 (b) and (c) exhibit accuracy on both the forget set and retain set, which can vary by up to 15 percentage points depending on the hyperparameters. Second, finding a balance between reducing accuracy on the forget set while maintaining accuracy on the retain set is challenging. As shown in Figure 1 (a), improving performance on the retain set tends to result in a clear decrease in performance on the forget set, and vice versa.

We argue overfitting the fine-tuned model to the forget set greatly diminishes performance on the retain set when unlearning is applied; conversely, underfitting the model to the forget set leads to ineffective unlearning, where the forgetting performance does not decrease sufficiently. Empirical evidence supporting our claim is presented in Section 4.3. Additionally, for successful unlearning,

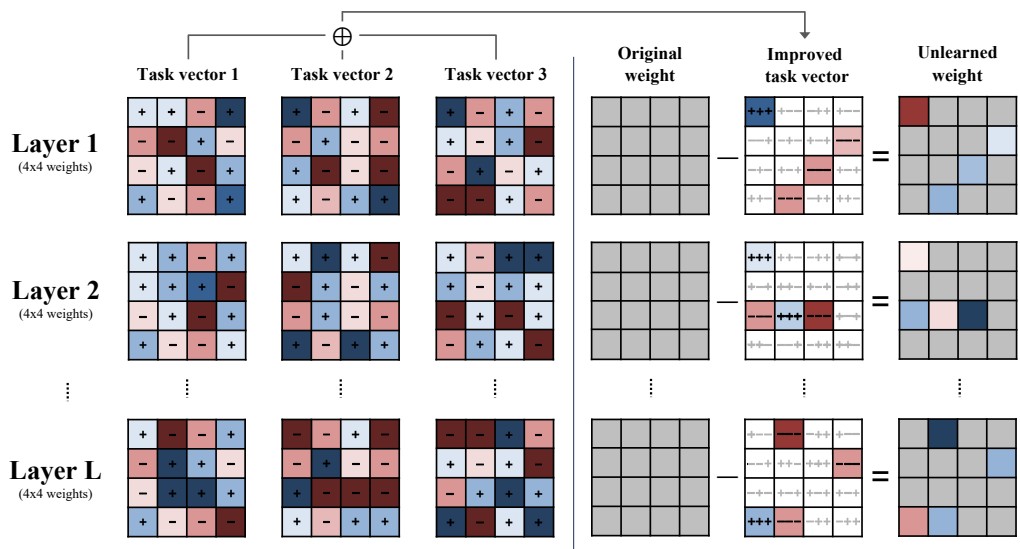

Figure 2: **Illustration of the proposed method.** Our `NegMerge` enhances task arithmetic by computing an improved task vector. Specifically, 1) multiple task vectors derived from fine-tuned models trained with different hyperparameters are utilized. 2) we compute the improved task vector by merging (⊕) only the elements that retain a consistent sign across task vectors while masking elements with differing signs to 0. 3) this refined task vector is used for negation from the original weights. The color intensity in the cells reflects the magnitude of the task vector elements; darker blue represents larger positive values, lighter blue indicates smaller positives, while darker red represents larger negative values, and lighter red indicates smaller negatives.

the task vector should exclusively represent the information to be forgotten within the framework of negation. This requires that the fine-tuned model precisely fits the forget set while it preserves the original knowledge.

Achieving both is challenging due to the nature of fine-tuning, where the only forget set is accessible; a model that fits the forget set well will inevitably lose knowledge of the retain set, and vice versa (Kirkpatrick et al., 2017). As we observed, this makes it difficult to achieve the desired balance with only a single model, which leads us to consider that aggregating multiple models could suggest more effective unlearning. However, we also observed that conventional model merging methods like Uniform Merging, Greedy Merging, TIES-Merging, and MagMax, which do not account for the characteristics of machine unlearning, fail this trade-off as displayed in Figure 1 (a). In contrast, our method, specialized in unlearning, surpasses this trade-off and achieves superior performance. We aim to use the given fine-tuned models more effectively and enhance unlearning outcomes while avoiding additional training costs. More details will be provided in Section 3.2.

**Our Unlearning scenarios.** In our study, we explore two distinct unlearning scenarios. The first scenario is the one described above, where a vision-language model like CLIP (Radford et al., 2021) is made to forget the knowledge of a specific class. For this scenario, we adopt the evaluation protocol for unlearning proposed in the original paper (Ilharco et al., 2022a). The other scenario involves a standard image classification network like ResNet (He et al., 2016) trained using cross-entropy loss on images and class labels. In this case, the model is made to forget the knowledge of specific training data. Here, we calculate the task vectors by fine-tuning the model only using the forget set: $\theta_{unlearn} = \theta_{ori} - \lambda(\theta_{ft}^{forget} - \theta_{ori})$ for both scenarios.

## 3.2 NEGMERGE: IMPROVED TASK ARITHMETIC FOR MACHINE UNLEARNING

Given multiple models fine-tuned on the forget set, which applied various training configurations to ensure diversity among the fine-tuned models, we propose a method that neatly aggregates the model for effective unlearning. Our proposed method called `NegMerge` consists of the following steps: 1) We calculate the task vectors using all the fine-tuned models, 2) We identify the elements corresponding to the forget set in each vector, and 3) Finally, we compute the final task vector by using the identified elements, and perform machine unlearning by subtracting this final task

vector from the original model. We provide a detailed description of each step below, and Figure 2 illustrates the overview of our method.

**Preparing Diverse Fine-Tuned Models.** There are numerous methods for preparing diverse fine-tuned models on the forget set. A simple yet effective approach is just altering hyperparameters such as learning rate and the number of epochs or employing data augmentation techniques like RandAugment (Cubuk et al., 2020) and CutMix (Yun et al., 2019). In this work, we focus on making minimal adjustments to the existing training setup, either by modifying RandAugment parameters or adjusting training configurations like the number of epochs. Further details on these adjustments can be found in Section 4.1. While additional techniques could further enhance model diversity and improve unlearning performance, these are left for future exploration.

**Identifying Elements in the Task Vector Corresponding to the Forget Set.** We derive task vectors from the fine-tuned models and analyze them to determine which elements (in weights) correspond to the forget set. We conjecture that elements that consistently show the same sign across task vectors are attributed to the forget set, as each model is trained to align with this set, regardless of the training configurations. On the other hand, components that exhibit differing signs are considered less related to the forget set, as their variations are more likely a result of different training configurations rather than supervision from the forget set. Our conjecture regarding sign conflicts is supported by the unlearning performance reported in Table 4 and qualitative results visualized in Figure 3.

**Final Task Vector for Negation.** We compute the final task vector using the following formulation:

$$\tau_{\text{merged}} = \left( \frac{1}{n} \sum_{k=1}^{n} \tau_k \right) \odot \mathbf{1}_{\text{signs are equal}}, \tag{1}$$

where $n$ is the number of task vectors, $\odot$ denotes the Hadamard product (element-wise multiplication), and the vector $\mathbf{1}_{\text{signs are equal}}$ acts like a filter, containing 1 for elements where the signs of the corresponding components across all task vectors $\tau_k$ are the same and 0 where the signs differ[1]. As a result, only the components with consistent signs across all task vectors contribute to the final task vector, while those with differing signs are excluded by being set to zero. We then perform machine unlearning by negating this final task vector to the original model (Ilharco et al., 2022a).

**Computational Complexity.** The standard setup for task negation-based methods (Ilharco et al., 2022a; Ortiz-Jimenez et al., 2024) typically involves conducting multiple evaluations (*i.e.*, usually done 20 iterations for the merging coefficient to find the optimal coefficient $\lambda \in \{0.0, 0.05, \ldots, 1.0\}$). For vanilla task arithmetic, this requires evaluating the coefficients $m$ times ($m = 20$ in the papers (Ilharco et al., 2022a; Ortiz-Jimenez et al., 2024)) for $n$ models, leading to a significant computational cost of $O(mn)$. In contrast, our merging method requires only $m$ evaluations for a single merged task vector, making it a more computationally efficient approach with a cost of $O(m)$. Therefore, while Task Arithmetic uses a single model in its final stage, achieving optimal performance demands more computation than our approach indeed. We believe this highlights the advantages of our method over the competing methods.

**Relationship with TIES-merging.** We highlight some explicit differences compared to a similar-looking strong method TIES-merging (Yadav et al., 2024). Since TIES-merging also performs selective merging based on signs, unlike our method, it includes elements with inconsistent signs across task vectors in the merging process. Specifically, it sums the values of these elements, checks the sign of the result, and sets the values of elements that do not match this sign to zero. The final task vector is obtained by merging these adjusted task vectors.

However, we argue that elements with inconsistent signs across task vectors are more closely related to the retain set than the forget set. Therefore, by including these sign-inconsistent elements in the merging process, TIES-merging may alter the knowledge of the retain set in the original model. For effective unlearning, task arithmetic should minimally affect the original model's knowledge of the retain set, but TIES-merging could significantly change this knowledge, making it unsuitable for machine unlearning. The results presented in Table 1 along with additional empirical evidence provided in Section 4.3, support our claim,

---

[1]This operation is based on sign unanimity and could be adjusted with additional hyperparameters to allow partial consensus, we opt for a simpler approach.

# 4 EXPERIMENTS

## 4.1 EXPERIMENTAL SETUPS

**Datasets and Backbones.** In the CLIP scenario (*i.e.*, referred to as the scenario using a vision-language model), we follow the training and evaluation protocols of Ilharco et al. (2022a). We assess unlearning performance on eight datasets: SUN397 (Xiao et al., 2016), Cars (Krause et al., 2013), RESISC45 (Cheng et al., 2017), EuroSAT (Helber et al., 2019), SVHN (Yuval, 2011), GT-SRB (Stallkamp et al., 2011), MNIST (LeCun, 1998), and DTD (Cimpoi et al., 2014). We use the pre-trained CLIP ViT-{B/32, B/16, L/14} models (Radford et al., 2021) for these experiments. In the standard classifier scenario, we evaluate unlearning performance on CIFAR-10 (Krizhevsky et al., 2009) using a ResNet-18 (He et al., 2016) model.

**Baselines and Metrics.** For the CLIP scenario, we compare our method with five existing methods: Task Arithmetic (Ilharco et al., 2022a), Uniform Merge (Wortsman et al., 2022), Greedy Merge (Wortsman et al., 2022), TIES-Merging (Yadav et al., 2024), and MagMax (Marczak et al., 2024). For the Greedy Merge, we rank models by their loss on the retain set and merge them in a direction that minimizes this loss. We evaluate performance by measuring accuracy on the forget set $D_f$ and the retain set $D_r$.

In the standard classifier scenario, we follow Fan et al. (2023) to compare our method against eight unlearning techniques: Fine-tuning (Warnecke et al., 2021), Random Labeling (Golatkar et al., 2020), Gradient Ascent (Thudi et al., 2022), Influence Unlearning (Koh & Liang, 2017), $\ell$1-sparse (Jia et al., 2023), Boundary Shrink and Expand (Chen et al., 2023), and SalUn (Fan et al., 2023). We also compare against Task Arithmetic (Ilharco et al., 2022a), Uniform Merge (Wortsman et al., 2022), TIES-Merging (Yadav et al., 2024), and MagMax (Marczak et al., 2024). The objective is to match the unlearned model's performance to that of a fully retrained model. Greedy Merge (Wortsman et al., 2022) is infeasible for comparison in this scenario, only using the forget set. We use the accuracies of the retain set $D_r$, forget set $D_f$, and test set $D_{test}$ to evaluate performance. To assess privacy protection, we employ the Membership Inference Attack (MIA) metric (Carlini et al., 2022), aiming to achieve similar results to the fully retrained model.

**Implementation Details.** In the CLIP scenario, for fine-tuning, we set the batch size to 128 and use a learning rate of 1e-5 with a cosine annealing schedule. We utilize the AdamW optimizer, applying a weight decay of 0.1. We enhance the diversity of the fine-tuned models by adjusting the configurations of RandAugment. Specifically, we vary the number of sequential augmentation transformations (ranging from 1 to 3) and the magnitude of these transformations (ranging from 1 to 10). A total of 30 models are fine-tuned. Unlike previous works, we incorporate data augmentation directly into the fine-tuning process, which requires adjusting the number of training epochs to better accommodate the augmented data. Consequently, the number of training epochs is set as follows: 70 epochs for Cars, 100 epochs for DTD, 40 epochs for EuroSAT, GTSRB, RESISC45, SUN397, and 30 epochs for MNIST and SVHN.

In the standard image classifier unlearning scenario, for the CIFAR-10 dataset, we set the batch size to 256 and the learning rate to 0.05. Since CIFAR-10 has relatively low image quality, instead of applying data augmentation, we vary the training hyperparameters. We set the number of epochs to 40, 50, and 60, the weight decay to 0.0001, 0.00005, and 0.00001, and the label smoothing to 0, 0.05, and 0.1 to enhance the diversity of the fine-tuned models. The total number of models used in the model merge is 27.

## 4.2 EXPERIMENTAL RESULTS

**CLIP Unlearning Scenario.** Table 1 presents the evaluation results across three variants of the CLIP model (ViT-B/32, ViT-B/16, and ViT-L/14) in the CLIP unlearning scenario. We follow the setup from Task Arithmetic (Ilharco et al., 2022b), freezing the final classification layer of CLIP's text encoder during fine-tuning. With the observation that freezing the classification layer does not affect accuracy (Ilharco et al., 2022b), we do not consider unfreezing the final layer of CLIP's text encoder in CLIP unlearning scenario. Our method achieves the best reduction in accuracy on the forget set $D_f$ across all backbone models, which demonstrates its generalizability regardless of model size and architecture.

Table 1: **Unlearning Performance on CLIP ViT Models.** Results are shown for CLIP ViT-{B/32, B/16, L/14}, reporting average accuracy (%) on the eight target tasks we wish to forget (Cars, DTD, EuroSAT, GTSRB, MNIST, RESISC45, SUN397, and SVHN), and the control task to remain (ImageNet). We compare our method with Task Arithmetic (Ilharco et al., 2022a), Linear Task Arithmetic (Ortiz-Jimenez et al., 2024), Uniform Merge (Wortsman et al., 2022), Greedy Merge (Wortsman et al., 2022), TIES-Merging (Yadav et al., 2024), and MagMax (Marczak et al., 2024). * indicates that the numbers are borrowed from the original papers. † denotes the best results achieved through hyperparameter search. ‡ combines models in descending order of losses. Time denotes the merging time, measured in seconds, taken to merge 30 models on the Cars dataset using CLIP ViT-B/32, which is averaged over three runs.

| Method | ViT-B/32 | | ViT-B/16 | | ViT-L/14 | | Time (sec) |
|---|---|---|---|---|---|---|---|
| | Acc $D_f(\downarrow)$ | Acc $D_r(\uparrow)$ | Acc $D_f(\downarrow)$ | Acc $D_r(\uparrow)$ | Acc $D_f(\downarrow)$ | Acc $D_r(\uparrow)$ | |
| Pre-trained | 48.13 | 63.33 | 55.49 | 68.32 | 65.19 | 75.54 | - |
| Task Arithmetic | | | | | | | |
| *Paper number** | 24.00 | 60.90 | 21.30 | 65.40 | 19.00 | 72.90 | - |
| Single Best Model† | 23.63 | 60.60 | 20.64 | 64.04 | 19.17 | 72.09 | - |
| Uniform Merge | 22.50 | 60.55 | 21.51 | 64.60 | 18.10 | 71.91 | $12_{\pm0.1}$ |
| Greedy Merge‡ | 23.31 | 60.75 | 21.34 | 64.54 | 17.71 | 71.99 | $607_{\pm2.6}$ |
| TIES-Merging | 26.21 | 61.08 | 23.78 | 64.72 | 22.70 | 72.41 | $128_{\pm10.1}$ |
| MagMax | 25.24 | 60.95 | 24.45 | 64.78 | 21.71 | 72.55 | $24_{\pm1.8}$ |
| **NegMerge (Ours)** | **20.76** | 60.36 | **19.24** | 64.54 | **17.32** | 72.08 | $37_{\pm1.2}$ |
| Linear Task Arithmetic | | | | | | | |
| *Paper number** | 10.90 | 60.80 | 11.30 | 64.80 | - | - | - |
| Single Best Model† | 8.88 | 60.16 | 6.92 | 64.62 | - | - | - |
| Uniform Merge | 9.12 | 60.47 | 6.84 | 65.26 | - | - | $19_{\pm2.3}$ |
| Greedy Merge‡ | 8.73 | 60.27 | 6.80 | 64.72 | - | - | $1696_{\pm35.3}$ |
| TIES-Merging | 10.66 | 60.38 | 8.44 | 65.12 | - | - | $378_{\pm8.0}$ |
| MagMax | 11.33 | 60.67 | 8.65 | 65.17 | - | - | $164_{\pm2.4}$ |
| **NegMerge (Ours)** | **8.03** | 60.58 | **6.60** | 65.40 | - | - | $194_{\pm1.6}$ |

For CLIP ViT-B/32, our method reduces the accuracy on the forget set $D_f$ to 20.76%. This out-performs Task Arithmetic (23.63%), Uniform Merge (22.50%), and Greedy Merge (23.31%). The accuracies on the retain set $D_r$ for all methods are around 60%. This is because we configure the model to ensure it does not fall below 95% of the pre-trained model's original accuracy (66.66%) on the validation set. Given this controlled performance on the retain set, it is appropriate to compare the effectiveness of techniques based solely on their forget set performance. This follows the setup from the original paper (Ilharco et al., 2022a). Therefore, the superior reduction in accuracy on forget set $D_f$ highlights the effectiveness of our method. Our method continues to show strong results using CLIP ViT-B/16, reducing the accuracy on the forget set to 19.24%, which outperforms Task Arithmetic (20.64%). For the CLIP ViT-L/14 model, our method also achieves the best performance on forget set, reducing it to 17.32%. In contrast, MagMax and TIES-Merging show worse results in terms of accuracy on the forget set. Regarding required merging time, our method spends slightly more time than Uniform Merge and MagMax but is far more effective. Additionally, Greedy Merge Wortsman et al. (2022) and TIES-Merging Yadav et al. (2024) are significantly slower than our method, and our approach outperforms them by a large margin in terms of accuracy.

To provide a more comprehensive evaluation of our method, we employ *linear task arithmetic*, where Neural Tangent Kernel (NTK) (Ortiz-Jimenez et al., 2024) is applied to the *standard task arithmetic* (Ilharco et al., 2022a). The experimental results are presented in the lower part of Table 1, where we conduct evaluations using the CLIP ViT-B/32 and ViT-B/16 backbones. Due to computational resource constraints, we are unable to include results for ViT-L/14. Our method achieves the best unlearning performance, while the second-best method, Greedy Merge, requires significantly more time for merging (1696.5 and 194.2, respectively).

**Standard Classifier Unlearning Scenario.** Table 2 presents a comparison of various unlearning techniques for random data forgetting on CIFAR-10 using ResNet-18. In this task, 10% of the training set is randomly selected, and the goal is to make the model forget the knowledge associated with this subset while maintaining its performance on the retain set. The fully retrained model serves as the ideal benchmark for both forget, retain and privacy tasks. The "Avg. Gap" metric is critical

Table 2: **Unlearning Performance for 10% Random Data Forgetting on CIFAR-10 using ResNet-18.** The results are expressed as a±b, representing the mean (a) and standard deviation (b) across three independent trials. The Avg. Gap is computed as the average of the performance differences observed in various accuracy-related metrics, including Acc $D_r$, Acc $D_f$, Acc $D_{test}$, and MIA. These metrics are favorable when they are close to the performance of the *Retrain model* ($\simeq$). * indicates that the numbers are borrowed from Fan et al. (2023). † denotes the best results achieved through hyperparameter search.

| Methods | Used Splits | Acc $D_r(\simeq)$ | Acc $D_f(\simeq)$ | Acc $D_{test}(\simeq)$ | MIA($\simeq$) | Avg. Gap($\downarrow$) |
|---|---|---|---|---|---|---|
| **Retrain *** | retain | $100.00_{\pm0.00}$ | $94.76_{\pm0.69}$ | $94.26_{\pm0.02}$ | $12.88_{\pm0.09}$ | 0.00 |
| Random Labeling * | | $99.67_{\pm0.14}$ | $92.39_{\pm0.31}$ | $92.83_{\pm0.38}$ | $37.36_{\pm0.06}$ | 7.15 |
| Influence * | all | $99.20_{\pm0.22}$ | $98.93_{\pm0.28}$ | $93.20_{\pm1.03}$ | $2.67_{\pm0.01}$ | 4.06 |
| SalUn * | | $99.62_{\pm0.12}$ | $97.15_{\pm0.43}$ | $93.93_{\pm0.29}$ | $14.39_{\pm0.82}$ | 1.15 |
| Finetune * | retain | $99.88_{\pm0.08}$ | $99.37_{\pm0.55}$ | $94.06_{\pm0.27}$ | $2.70_{\pm0.01}$ | 3.78 |
| $\ell$1-sparse * | | $97.74_{\pm0.33}$ | $95.81_{\pm0.62}$ | $91.59_{\pm0.57}$ | $9.84_{\pm0.00}$ | 2.26 |
| Gradient Ascent * | | $99.50_{\pm0.38}$ | $99.31_{\pm0.54}$ | $94.01_{\pm0.47}$ | $1.70_{\pm0.01}$ | 4.12 |
| Boundary Shrink * | | $98.29_{\pm2.50}$ | $98.22_{\pm2.52}$ | $92.69_{\pm2.99}$ | $8.96_{\pm0.13}$ | 2.67 |
| Boundary Expanding * | forget | $99.42_{\pm0.33}$ | $99.41_{\pm0.30}$ | $93.85_{\pm1.02}$ | $7.47_{\pm1.15}$ | 2.76 |
| Random Labeling | | $99.99_{\pm0.00}$ | $99.98_{\pm0.02}$ | $95.04_{\pm0.11}$ | $2.15_{\pm1.94}$ | 4.19 |
| SalUn | | $99.88_{\pm0.04}$ | $99.89_{\pm0.04}$ | $94.42_{\pm0.05}$ | $9.51_{\pm2.07}$ | 2.20 |
| Task Arithmetic | | | | | | |
| Single Best Model† | | $98.36_{\pm0.51}$ | $94.85_{\pm0.16}$ | $91.49_{\pm0.80}$ | $10.91_{\pm0.72}$ | 1.62 |
| Uniform Merge | | $98.70_{\pm0.91}$ | $95.83_{\pm2.17}$ | $92.36_{\pm1.16}$ | $10.14_{\pm2.93}$ | 1.75 |
| TIES-Merging | forget | $98.38_{\pm0.17}$ | $95.45_{\pm0.32}$ | $92.23_{\pm0.14}$ | $9.36_{\pm0.31}$ | 1.96 |
| MagMax | | $98.38_{\pm0.12}$ | $97.97_{\pm0.77}$ | $91.53_{\pm0.00}$ | $8.45_{\pm2.60}$ | 3.00 |
| **NegMerge (Ours)** | | $99.15_{\pm0.24}$ | $96.63_{\pm0.59}$ | $92.71_{\pm0.39}$ | $12.87_{\pm1.29}$ | **1.07** |

in evaluating how closely each method replicates the performance of the retrained model across key metrics such as $D_r$ (accuracy on retain set), $D_f$ (accuracy on forget set), $D_{test}$ (accuracy on test set), and the MIA score.

Our method achieves an average gap of 1.07, indicating minimal performance degradation and demonstrating that it effectively unlearns specific information while preserving the model's overall capabilities. SalUn, which uses all data splits for unlearning, achieves an average gap of 1.15, similar to the retrained model. However, our method, which only relies on the forget set, outperforms it with an average gap of 1.07, indicating our approach's efficiency in retaining generalization without relying on the retain set. Task Arithmetic and merging methods, including Uniform Merge, TIES-Merging, and MagMax, result in larger gaps (1.62, 1.75, 1.96, and 3.00, respectively), highlighting that our method achieves a better balance between forgetting and preserving knowledge in the retain set.

Overall, our method stands out by maintaining performance close to the retrained model, particularly in preserving accuracy on $D_r$ and $D_{test}$, while effectively reducing accuracy on $D_f$. Additionally, our method offers strong privacy protection, with an MIA score of 12.87, nearly identical to that of the retrained model (12.88), ensuring that the model forgets the targeted data without introducing privacy vulnerabilities.

## 4.3 EMPIRICAL ANALYSES

**Regarding Our Key Assumptions.** Our method relies on two key assumptions: (1) Effective unlearning requires the fine-tuned model to maintain high performance on the forget set without degrading performance on the retain set, and (2) To accomplish this, only elements with consistent signs across task vectors should be used during the merging process.

Table 3 presents the evaluation results on both the forget set and retain set for the models derived by *adding* task vector to the original model. According to the results, most merging methods exhibit high performance on the forget set $D_f$. However, we observe that, except for our method, the performance on the retain set $D_r$ significantly drops. Given that our unlearning method achieves the highest performance, this supports our first assumption that high performance on the forget set is necessary while maintaining performance on the retain set. Additionally, unlike our method,

Table 3: **Comparative Performance on Different Datasets.** Finetuning results showing the unlearning accuracy $D_f$ and remaining accuracy $D_r$ across various methods on Cars, DTD, and SUN397 datasets. Finetuning models derived by *adding* task vector to the original model. $*$ denotes our reproduced results based on the configurations from Ilharco et al. (2022a). $\dagger$ represents the best results achieved through hyperparameter tuning, including adjustments to data augmentation. $\ddagger$ combines models in descending order of losses. A higher $D_f$ is better ($\uparrow$), while $D_r$ is preferable when it is close to the performance of the $D_r$ of the pre-trained model ($\simeq$).

| Method | Cars | | DTD | | SUN397 | |
|---|---|---|---|---|---|---|
| | Acc $D_f(\uparrow)$ | Acc $D_r(\simeq)$ | Acc $D_f(\uparrow)$ | Acc $D_r(\simeq)$ | Acc $D_f(\uparrow)$ | Acc $D_r(\simeq)$ |
| Pretrained | 59.6 | **66.7** | 43.9 | **66.7** | 63.3 | **66.7** |
| Task Arithmetic | | | | | | |
| *Paper config*$^*$ | 85.0 | 58.6 | 78.7 | 49.3 | 74.9 | 59.8 |
| Single Best Model$^\dagger$ | 86.6 | 52.7 | 76.9 | 48.4 | 76.5 | 55.7 |
| Uniform Merge | 87.2 | 55.3 | 79.0 | 52.8 | 76.0 | 57.1 |
| Greedy Merge$^\ddagger$ | 87.5 | 55.2 | 79.3 | 52.8 | 76.2 | 57.1 |
| TIES-Merging | 85.3 | 34.7 | 75.2 | 25.4 | 71.4 | 44.3 |
| MagMax | 66.8 | 5.8 | 59.3 | 2.8 | 52.0 | 17.9 |
| **NegMerge (Ours)** | 87.1 | **61.7** | 76.3 | **63.0** | 76.3 | **63.4** |

Uniform Merge, which merges all elements, leads to a substantial performance drop on the retain set. This observation supports our second assumption that only sign-consistent elements should be involved in the merging process. These results demonstrate the effectiveness of our approach in addressing the trade-off between the forget and retain sets, underscoring the rationale behind the design choices in our method.

Our experimental results consistently show that TIES-Merging underperforms. The primary reason for this is the significant drop in retain set performance, as shown in Table 3. The performance of TIES-Merging on $D_r$ is lower than that of basic methods like Uniform Merge, and we believe this low $D_r$ performance is the main factor behind TIES-Merging's poor unlearning performance. A similar trend is observed with MagMax. These results reinforce our discussion in Section 3.2, where we argue that the design choices in TIES-Merging are ineffective at preserving knowledge of the retain set, leading to lower unlearning performance.

Table 4: **Impact of Sign Conflict in Weights for Unlearning.** The results present unlearning performance across various datasets, comparing three different methods. "All," Uniform Merge, uses all indices without regard to sign conflict, "Conflict" uses only indices with conflicting signs, and "Non-conflict," our proposed method, uses only indices with consistent signs across task vectors.

| Method | Cars | | DTD | | EuroSAT | | SUN397 | |
|---|---|---|---|---|---|---|---|---|
| | Acc $D_f(\downarrow)$ | Acc $D_r(\uparrow)$ | Acc $D_f(\downarrow)$ | Acc $D_r(\uparrow)$ | Acc $D_f(\downarrow)$ | Acc $D_r(\uparrow)$ | Acc $D_f(\downarrow)$ | Acc $D_r(\uparrow)$ |
| All | 31.7 | 60.4 | 29.6 | 60.6 | 8.9 | 60.8 | 51.4 | 60.5 |
| Conflict | 40.2 | 60.2 | 31.9 | 60.3 | 11.1 | 60.7 | 58.3 | 60.9 |
| **Non-conflict** | 27.4 | 60.4 | 27.2 | 60.5 | 7.9 | 60.2 | 47.2 | 60.6 |

**Effect of Sign Conflict on Unlearning Performance.** We argue that elements with consistent signs across multiple task vectors correspond to knowledge related to the forget set, while elements with conflicting signs are less relevant to the forget set.

To verify this, we compare unlearning performance when our method is applied in reverse. The experimental results are shown in Table 4. We use the CLIP ViT-B/32 model and the standard task arithmetic. The *All* method refers to the Uniform Merge approach, which uses all elements without considering sign conflicts. The *Conflict* method uses only elements with conflicting signs, while our proposed *Non-conflict* method uses only elements with consistent signs. The results show that the *Conflict* method significantly degrades unlearning performance, while the *All* method performs better than *Conflict* but is outperformed by our *Non-conflict* method. These experimental results indicate that the design choice of merging only sign-consistent elements is effective.

In Figure 3, we demonstrate the effectiveness of our method using Grad-CAM visualizations on the RESISC45 dataset. We compare the *Conflict* and our *Non-conflict* methods, and include visualiza-

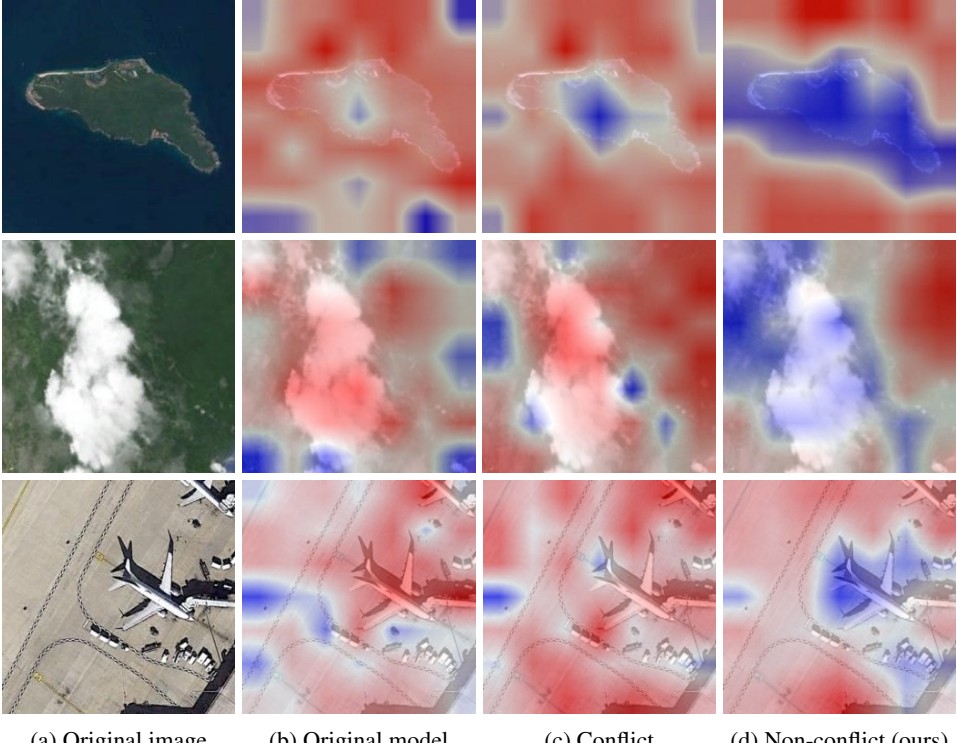

(a) Original image     (b) Original model     (c) Conflict     (d) Non-conflict (ours)

Figure 3: **Impact of Sign Conflicts on Unlearning.** We illustrate the effectiveness of our technique in terms of the sign conflicts using Grad-CAM on the RESISC45 dataset. The first row represents the *island* class, the second row corresponds to the *cloud* class, and the third row represents the *airplane* class. Red areas indicate regions of high relevance or activation in response to the class label, which is crucial for the model's decision-making process. Blue areas represent regions of low relevance or activation, which have little to no influence on the model's output for the corresponding class.

tions of the original model as a baseline. The red areas represent regions where the model strongly associates with the class label, while the blue areas indicate regions with less relevance. In the first row (island class), we observe that the *Conflict* method directs the model's attention to the island's location, resembling the behavior of the original model. In contrast, our method does not highlight the island's area, which suggests that the model has successfully forgotten its knowledge of the island. The same pattern appears in the second row for the cloud class and in the third row for the airplane class. These visual results clearly demonstrate that our proposed method, `NegMerge`, is more effective for unlearning.

## 5 CONCLUSION

In this paper, we propose a novel machine unlearning technique, `NegMerge`, based on task arithmetic and model merging. We hypothesize that multiple fine-tuned models are necessary for effective unlearning based on the observation of a trade-off between accuracy on the forget set and the remain set. Building on the fact that existing techniques generate numerous fine-tuned models through validation using various hyperparameters, we propose a method that utilizes all derived fine-tuned models. Assuming that elements with consistent signs across task vectors obtained from the fine-tuned models are related to the forget set, we merge only those elements. This approach enables us to compute task vectors that fit the forget set more effectively while preserving the knowledge in the retain set, thus overcoming the trade-off. We then perform forgetting by negation with the merged task vector. Our `NegMerge` is tested on the CLIP ViT models and the standard ResNet18 classifier, achieving new state-of-the-art performance across nine datasets.

**Limitations.** Limitation of this work is its reliance on empirical approaches without formal theoretical justification. In future research, we aim to validate our assumptions theoretically and develop an analytical solution informed by these insights.

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

# Appendix

This appendix comprises the following materials: 1) More experimental results (Section A), 2) Full charts of the CLIP unlearning scenario (Section B), 3) Ratio of zeroed elements in the merged vector (Section C), 4) Results in diverse model pool (Section D), 5) Memory and computational efficiency (Section E), and 6) Theoretical insights (Section F).

## A  FURTHER EXPERIMENTAL RESULTS

### A.1  FULL RESULTS

Tables A1, A2 and A3 show the full accuracy results for the eight tasks and the three CLIP models we examine. Tables A4 and A5 show the full accuracy results on Linear Task Arithmetic for the eight tasks and the two CLIP models we examine.

Table A1: **ViT-B/32 standard.** Results are shown for various methods across multiple datasets (Cars, DTD, EuroSAT, GTSRB, MNIST, RESISC45, SUN397, and SVHN).

| Method | Cars | | DTD | | EuroSAT | | GTSRB | | MNIST | | RESISC45 | | SUN397 | | SVHN | |
|---|---|---|---|---|---|---|---|---|---|---|---|---|---|---|---|---|
| | $D_f$ ($\downarrow$) | $D_r$ ($\uparrow$) | $D_f$ ($\downarrow$) | $D_r$ ($\uparrow$) | $D_f$ ($\downarrow$) | $D_r$ ($\uparrow$) | $D_f$ ($\downarrow$) | $D_r$ ($\uparrow$) | $D_f$ ($\downarrow$) | $D_r$ ($\uparrow$) | $D_f$ ($\downarrow$) | $D_r$ ($\uparrow$) | $D_f$ ($\downarrow$) | $D_r$ ($\uparrow$) | $D_f$ ($\downarrow$) | $D_r$ ($\uparrow$) |
| Task Arithmetic[†] | 29.0 | 59.9 | 30.4 | 60.8 | 10.4 | 60.9 | 9.1 | 60.9 | 21.2 | 60.6 | 30.7 | 60.8 | 50.6 | 59.9 | 7.6 | 60.9 |
| Uniform Merge | 31.7 | 60.4 | 29.6 | 60.6 | 8.9 | 60.8 | 7.0 | 60.0 | 20.5 | 61.4 | 23.8 | 60.1 | 51.4 | 60.5 | 7.3 | 60.7 |
| Greedy Merge[‡] | 31.0 | 60.3 | 29.5 | 60.6 | 9.4 | 60.8 | 8.4 | 60.5 | 21.3 | 62.0 | 28.3 | 60.7 | 51.4 | 60.4 | 7.2 | 60.7 |
| TIES-Merging | 34.0 | 60.3 | 33.1 | 61.3 | 11.6 | 61.1 | 10.2 | 61.3 | 26.1 | 62.4 | 33.4 | 61.0 | 53.8 | 60.3 | 7.5 | 60.9 |
| MagMax | 35.6 | 60.6 | 31.9 | 61.1 | 10.5 | 60.7 | 8.4 | 60.8 | 20.1 | 60.7 | 30.7 | 60.6 | 55.4 | 61.1 | 9.3 | 62.0 |
| **NegMerge (Ours)** | 27.4 | 60.4 | 27.2 | 60.5 | 7.9 | 60.2 | 6.2 | 60.0 | 20.5 | 59.9 | 22.6 | 60.5 | 47.2 | 60.6 | 7.2 | 60.9 |

Table A2: **ViT-B/16 standard.** Results are shown for various methods across multiple datasets (Cars, DTD, EuroSAT, GTSRB, MNIST, RESISC45, SUN397, and SVHN).

| Method | Cars | | DTD | | EuroSAT | | GTSRB | | MNIST | | RESISC45 | | SUN397 | | SVHN | |
|---|---|---|---|---|---|---|---|---|---|---|---|---|---|---|---|---|
| | $D_f$ ($\downarrow$) | $D_r$ ($\uparrow$) | $D_f$ ($\downarrow$) | $D_r$ ($\uparrow$) | $D_f$ ($\downarrow$) | $D_r$ ($\uparrow$) | $D_f$ ($\downarrow$) | $D_r$ ($\uparrow$) | $D_f$ ($\downarrow$) | $D_r$ ($\uparrow$) | $D_f$ ($\downarrow$) | $D_r$ ($\uparrow$) | $D_f$ ($\downarrow$) | $D_r$ ($\uparrow$) | $D_f$ ($\downarrow$) | $D_r$ ($\uparrow$) |
| Task Arithmetic[†] | 31.6 | 63.8 | 26.1 | 63.8 | 7.6 | 64.3 | 7.7 | 64.5 | 8.9 | 64.0 | 27.2 | 64.4 | 49.1 | 63.7 | 6.9 | 63.9 |
| Uniform Merge | 32.9 | 64.6 | 26.3 | 64.5 | 9.8 | 64.8 | 7.0 | 64.1 | 13.9 | 65.0 | 25.6 | 64.7 | 49.7 | 64.6 | 6.9 | 64.7 |
| Greedy Merge[‡] | 32.9 | 64.6 | 25.0 | 63.7 | 9.9 | 64.7 | 7.0 | 64.1 | 12.4 | 64.8 | 25.6 | 64.6 | 51.1 | 65.1 | 6.9 | 64.7 |
| TIES-Merging | 39.4 | 65.0 | 27.4 | 64.0 | 10.2 | 64.8 | 8.6 | 64.6 | 11.1 | 64.9 | 33.6 | 65.3 | 53.2 | 64.8 | 6.7 | 64.3 |
| MagMax | 38.4 | 64.8 | 26.6 | 63.9 | 10.2 | 65.0 | 9.0 | 64.9 | 14.6 | 64.4 | 36.6 | 66.0 | 53.5 | 65.0 | 6.7 | 64.3 |
| **NegMerge (Ours)** | 28.8 | 64.8 | 25.2 | 64.5 | 9.8 | 65.9 | 7.1 | 64.4 | 10.7 | 63.8 | 20.3 | 63.9 | 45.2 | 64.4 | 7.0 | 64.6 |

Table A3: **ViT-L/14 standard.** Results are shown for various methods across multiple datasets (Cars, DTD, EuroSAT, GTSRB, MNIST, RESISC45, SUN397, and SVHN).

| Method | Cars | | DTD | | EuroSAT | | GTSRB | | MNIST | | RESISC45 | | SUN397 | | SVHN | |
|---|---|---|---|---|---|---|---|---|---|---|---|---|---|---|---|---|
| | $D_f$ ($\downarrow$) | $D_r$ ($\uparrow$) | $D_f$ ($\downarrow$) | $D_r$ ($\uparrow$) | $D_f$ ($\downarrow$) | $D_r$ ($\uparrow$) | $D_f$ ($\downarrow$) | $D_r$ ($\uparrow$) | $D_f$ ($\downarrow$) | $D_r$ ($\uparrow$) | $D_f$ ($\downarrow$) | $D_r$ ($\uparrow$) | $D_f$ ($\downarrow$) | $D_r$ ($\uparrow$) | $D_f$ ($\downarrow$) | $D_r$ ($\uparrow$) |
| Task Arithmetic[†] | 34.6 | 72.2 | 24.7 | 71.3 | 5.4 | 72.5 | 3.0 | 71.6 | 10.3 | 73.6 | 17.0 | 71.7 | 51.6 | 71.9 | 6.7 | 71.9 |
| Uniform Merge | 29.1 | 71.8 | 23.5 | 71.4 | 8.2 | 72.1 | 3.1 | 71.5 | 9.9 | 72.4 | 13.9 | 71.5 | 50.5 | 72.3 | 6.7 | 72.2 |
| Greedy Merge[‡] | 28.2 | 71.5 | 23.9 | 71.5 | 7.3 | 73.0 | 3.1 | 71.7 | 9.9 | 72.8 | 11.5 | 71.0 | 51.1 | 72.3 | 6.8 | 72.1 |
| TIES-Merging | 48.2 | 73.1 | 25.5 | 71.5 | 9.2 | 72.4 | 4.1 | 72.6 | 10.3 | 73.0 | 21.0 | 72.0 | 56.6 | 72.8 | 6.8 | 71.9 |
| MagMax | 39.2 | 72.0 | 28.7 | 72.7 | 9.9 | 73.6 | 4.2 | 72.5 | 10.7 | 73.5 | 20.6 | 72.2 | 53.7 | 72.1 | 6.7 | 71.9 |
| **NegMerge (Ours)** | 32.7 | 71.9 | 23.9 | 71.9 | 9.1 | 72.1 | 2.8 | 71.3 | 10.9 | 73.6 | 8.8 | 70.9 | 43.6 | 72.1 | 6.8 | 72.8 |

Table A4: **ViT-B/32 linear.** Results are shown for various methods across multiple datasets (Cars, DTD, EuroSAT, GTSRB, MNIST, RESISC45, SUN397, and SVHN).

| Method | Cars | | DTD | | EuroSAT | | GTSRB | | MNIST | | RESISC45 | | SUN397 | | SVHN | |
|---|---|---|---|---|---|---|---|---|---|---|---|---|---|---|---|---|
| | $D_f$ (↓) | $D_r$ (↑) | $D_f$ (↓) | $D_r$ (↑) | $D_f$ (↓) | $D_r$ (↑) | $D_f$ (↓) | $D_r$ (↑) | $D_f$ (↓) | $D_r$ (↑) | $D_f$ (↓) | $D_r$ (↑) | $D_f$ (↓) | $D_r$ (↑) | $D_f$ (↓) | $D_r$ (↑) |
| Task Arithmetic[†] | 13.5 | 60.2 | 15.2 | 59.7 | 0.1 | 60.3 | 0.2 | 60.2 | 0.1 | 61.0 | 2.6 | 59.6 | 38.8 | 59.9 | 0.7 | 60.5 |
| Uniform Merge | 14.2 | 60.4 | 15.3 | 60.2 | 0.0 | 60.3 | 0.2 | 60.8 | 0.0 | 60.2 | 2.6 | 60.2 | 39.7 | 60.4 | 0.8 | 61.3 |
| Greedy Merge[‡] | 14.4 | 60.3 | 15.8 | 60.2 | 0.0 | 60.4 | 0.2 | 60.2 | 0.0 | 60.5 | 2.6 | 60.3 | 36.3 | 59.7 | 0.7 | 60.6 |
| TIES-Merging | 19.3 | 60.4 | 16.5 | 60.0 | 0.3 | 60.5 | 0.2 | 60.5 | 0.0 | 60.4 | 5.6 | 60.4 | 42.8 | 60.3 | 0.8 | 60.5 |
| MagMax | 22.6 | 61.0 | 16.5 | 60.1 | 0.2 | 60.6 | 0.2 | 60.8 | 0.1 | 61.5 | 4.2 | 60.1 | 46.1 | 60.9 | 0.7 | 60.4 |
| **NegMerge (Ours)** | 12.1 | 60.6 | 15.6 | 60.4 | 0.0 | 60.9 | 0.2 | 61.2 | 0.0 | 60.9 | 1.6 | 60.1 | 34.0 | 59.8 | 0.7 | 60.8 |

Table A5: **ViT-B/16 linear.** Results are shown for various methods across multiple datasets (Cars, DTD, EuroSAT, GTSRB, MNIST, RESISC45, SUN397, and SVHN).

| Method | Cars | | DTD | | EuroSAT | | GTSRB | | MNIST | | RESISC45 | | SUN397 | | SVHN | |
|---|---|---|---|---|---|---|---|---|---|---|---|---|---|---|---|---|
| | $D_f$ (↓) | $D_r$ (↑) | $D_f$ (↓) | $D_r$ (↑) | $D_f$ (↓) | $D_r$ (↑) | $D_f$ (↓) | $D_r$ (↑) | $D_f$ (↓) | $D_r$ (↑) | $D_f$ (↓) | $D_r$ (↑) | $D_f$ (↓) | $D_r$ (↑) | $D_f$ (↓) | $D_r$ (↑) |
| Task Arithmetic[†] | 5.3 | 64.4 | 10.2 | 63.8 | 0.0 | 64.8 | 0.0 | 64.5 | 0.1 | 67.0 | 2.0 | 63.6 | 37.4 | 64.7 | 0.5 | 64.1 |
| Uniform Merge | 5.0 | 64.7 | 10.1 | 64.2 | 0.1 | 66.0 | 0.0 | 66.0 | 0.1 | 67.4 | 1.6 | 64.0 | 37.5 | 65.0 | 0.4 | 64.8 |
| Greedy Merge[‡] | 5.0 | 64.8 | 10.3 | 64.1 | 0.0 | 64.5 | 0.0 | 64.5 | 0.1 | 66.8 | 1.5 | 63.9 | 37.1 | 65.0 | 0.4 | 64.2 |
| TIES-Merging | 7.4 | 64.3 | 11.7 | 64.6 | 0.1 | 65.3 | 0.0 | 65.4 | 0.1 | 66.9 | 4.1 | 64.4 | 43.8 | 65.7 | 0.4 | 64.3 |
| MagMax | 8.8 | 64.6 | 12.3 | 64.9 | 0.0 | 64.9 | 0.0 | 64.9 | 0.1 | 67.2 | 5.1 | 64.9 | 42.5 | 65.4 | 0.4 | 64.6 |
| **NegMerge (Ours)** | 6.6 | 65.9 | 10.4 | 64.5 | 0.0 | 65.9 | 0.0 | 66.0 | 0.1 | 66.9 | 1.1 | 64.5 | 34.1 | 64.7 | 0.5 | 64.8 |

Table A6 shows the Impact of Sign Conflict in Weights for Unlearning for the eight tasks.

Table A6: **Impact of Sign Conflict in Weights for Unlearning.** The results present unlearning performance across various datasets using CLIP ViT-B/32, comparing three different methods. "All," Uniform Merge, uses all indices without regard to sign conflict, "Conflict" uses only indices with conflicting signs, and "Non-conflict," our proposed method, uses only indices with consistent signs across task vectors.

| Method | Cars | | DTD | | EuroSAT | | GTSRB | | MNIST | | RESISC45 | | SUN397 | | SVHN | |
|---|---|---|---|---|---|---|---|---|---|---|---|---|---|---|---|---|
| | $D_f(\downarrow)$ | $D_r(\uparrow)$ | $D_f(\downarrow)$ | $D_r(\uparrow)$ | $D_f(\downarrow)$ | $D_r(\uparrow)$ | $D_f(\downarrow)$ | $D_r(\uparrow)$ | $D_f(\downarrow)$ | $D_r(\uparrow)$ | $D_f(\downarrow)$ | $D_r(\uparrow)$ | $D_f(\downarrow)$ | $D_r(\uparrow)$ | $D_f(\downarrow)$ | $D_r(\uparrow)$ |
| All | 31.7 | 60.4 | 29.6 | 60.6 | 8.9 | 60.8 | 7.0 | 60.0 | 20.5 | 61.4 | 23.8 | 60.1 | 51.4 | 60.5 | 7.3 | 60.7 |
| Conflict | 40.2 | 60.2 | 31.9 | 60.3 | 11.1 | 60.7 | 9.1 | 60.6 | 24.0 | 61.9 | 32.3 | 60.2 | 58.3 | 60.9 | 8.8 | 60.6 |
| **Non-conflict** | 27.4 | 60.4 | 27.2 | 60.5 | 7.9 | 60.2 | 6.2 | 60.0 | 20.5 | 59.9 | 22.6 | 60.5 | 47.2 | 60.6 | 7.2 | 60.9 |

## A.2 ABLATION STUDY

Table A7 compares ways to derive the improved final task vector $\tau_{\text{merged}}$. We found that our originally proposed averaging method performed the best. This is likely because averaging helps smooth out potential outliers in individual models during merging, resulting in a more stable and effective task vector.

Table A7: **Ablation Study to derive the Improved Final Task Vector.** `NegMerge` (min), `NegMerge` (max), and `NegMerge` (avg) represent merging minimum, maximum, and average of task vectors elements, respectively. The experimental results are obtained using CLIP ViT-B/32.

| Method | Cars | | DTD | | EuroSAT | | GTSRB | | MNIST | | RESISC45 | | SUN397 | | SVHN | |
|---|---|---|---|---|---|---|---|---|---|---|---|---|---|---|---|---|
| | $D_f(\downarrow)$ | $D_r(\uparrow)$ | $D_f(\downarrow)$ | $D_r(\uparrow)$ | $D_f(\downarrow)$ | $D_r(\uparrow)$ | $D_f(\downarrow)$ | $D_r(\uparrow)$ | $D_f(\downarrow)$ | $D_r(\uparrow)$ | $D_f(\downarrow)$ | $D_r(\uparrow)$ | $D_f(\downarrow)$ | $D_r(\uparrow)$ | $D_f(\downarrow)$ | $D_r(\uparrow)$ |
| Task Arithmetic[†] | 29.0 | 59.9 | 30.4 | 60.8 | 10.4 | 60.9 | 9.1 | 60.9 | 21.2 | 60.6 | 30.7 | 60.8 | 50.6 | 59.9 | 7.6 | 60.9 |
| NegMerge (min) | 26.5 | 59.9 | 27.9 | 60.6 | 10.3 | 60.8 | 8.3 | 60.7 | 25.2 | 61.0 | 20.1 | 60.0 | 46.8 | 60.2 | 8.2 | 61.0 |
| NegMerge (max) | 28.2 | 60.3 | 27.4 | 60.4 | 10.7 | 60.9 | 7.5 | 60.4 | 28.7 | 60.2 | 26.0 | 61.0 | 48.3 | 60.7 | 7.3 | 60.8 |
| **NegMerge (avg)** | 27.4 | 60.4 | 27.2 | 60.5 | 7.9 | 60.2 | 6.2 | 60.0 | 20.5 | 59.9 | 22.6 | 60.5 | 47.2 | 60.6 | 7.2 | 60.9 |

## A.3 ADDITIONAL STANDARD CLASSIFIER UNLEARNING SCENARIO RESULTS

Table A8 presents a comparison of various unlearning techniques for 10% random data forgetting on CUB (Wah et al., 2011) using ResNet-18. This experiment is important because we can validate our method's effectiveness on fine-grained image classification. Table A9 presents a comparison of various unlearning techniques for 10% random data forgetting on CIFAR-10 using VGG-16. Only using the forget set `NegMerge` can effectively unlearn Df. These results show that `NegMerge` generalize across a wider range of datasets and model architectures.

Table A8: **Unlearning Performance for 10% Random Data Forgetting on CUB using ResNet-18.** The Avg. Gap is computed as the average of the performance differences observed in various accuracy-related metrics, including Acc $D_r$, Acc $D_f$, Acc $D_{test}$, and MIA. These metrics are favorable when they are close to the performance of the *Retrain model* ($\simeq$).

| Methods | Used Splits | Acc $D_r(\simeq)$ | Acc $D_f(\simeq)$ | Acc $D_{test}(\simeq)$ | MIA($\simeq$) | Avg. Gap($\downarrow$) |
|---|---|---|---|---|---|---|
| **Retrain** | retain | 78.55 | 56.43 | 74.61 | 80.47 | 0.00 |
| Gradient Ascent | | 66.75 | 57.26 | 66.60 | 67.61 | 8.38 |
| Boundary Shrink | | 66.88 | 61.60 | 64.14 | 100.00 | 11.71 |
| Boundary Expanding | forget | 65.32 | 61.60 | 58.80 | 73.62 | 10.27 |
| Random Labeling | | 64.13 | 57.43 | 59.54 | 71.79 | 9.79 |
| SalUn | | 66.69 | 59.60 | 63.88 | 74.46 | 7.94 |
| Task Arithmetic | | | | | | |
| Single Best Model[†] | forget | 74.68 | 58.60 | 70.56 | 100.00 | 7.41 |
| Uniform Merge | | 73.94 | 56.93 | 69.78 | 100.00 | 7.37 |
| **NegMerge (Ours)** | | 74.64 | 58.26 | 70.69 | 100.00 | **7.30** |

Table A9: **Unlearning Performance for 10% Random Data Forgetting on CIFAR-10 using VGG-16.** The Avg. Gap is computed as the average of the performance differences observed in various accuracy-related metrics, including Acc $D_r$, Acc $D_f$, Acc $D_{test}$, and MIA. These metrics are favorable when they are close to the performance of the *Retrain model* ($\simeq$). [*] indicates that the numbers are borrowed from Fan et al. (2023). [†] denotes the best results achieved through hyperparameter search.

| Methods | Used Splits | Acc $D_r(\simeq)$ | Acc $D_f(\simeq)$ | Acc $D_{test}(\simeq)$ | MIA($\simeq$) | Avg. Gap($\downarrow$) |
|---|---|---|---|---|---|---|
| **Retrain** [*] | retain | 99.99 | 94.02 | 93.06 | 10.36 | 0.00 |
| Random Labeling [*] | | 99.65 | 94.29 | 92.29 | 15.98 | 1.75 |
| Influence [*] | all | 98.78 | 98.33 | 91.69 | 2.71 | 3.63 |
| SalUn [*] | | 98.74 | 96.11 | 91.62 | 9.96 | 1.29 |
| Finetune [*] | retain | 99.54 | 98.49 | 92.64 | 3.76 | 2.98 |
| $\ell$1-sparse [*] | | 97.03 | 95.02 | 90.15 | 9.69 | 1.88 |
| Gradient Ascent [*] | | 99.37 | 99.07 | 93.63 | 1.36 | 3.81 |
| Boundary Shrink [*] | forget | 99.40 | 99.20 | 93.68 | 1.38 | 3.84 |
| Boundary Expanding [*] | | 99.39 | 99.20 | 93.68 | 1.42 | 3.84 |
| Task Arithmetic | | | | | | |
| Single Best Model[†] | forget | 97.26 | 94.90 | 90.10 | 10.34 | 1.64 |
| **NegMerge (Ours)** | | 98.00 | 95.74 | 91.01 | 10.10 | **1.50** |

# B    FULL CHARTS OF CLIP UNLEARNING SCENARIO

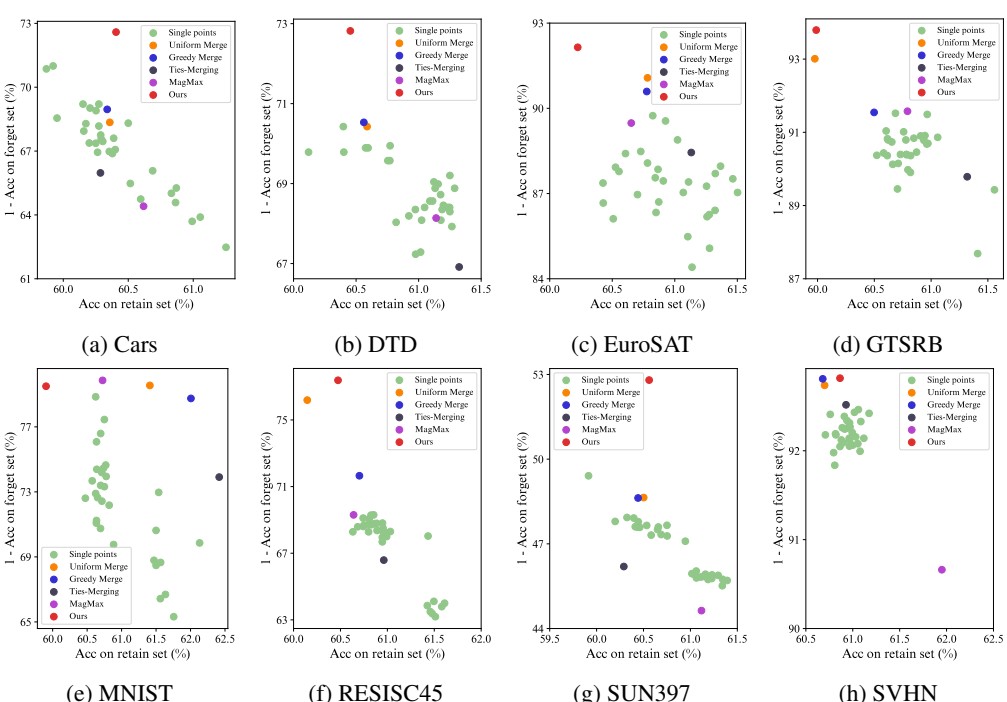

Figure B1: **Comparison of Merged Models on ViT-B/32.** Performance metrics for merged models showing accuracy on retain set and forget set across different models. Methods positioned towards the upper right corner are generally considered to be better performers.

# C    RATIO OF ZEROED ELEMENTS IN THE MERGED VECTOR

Tables C1 and C2 present the proportion of zeroed weight elements based on the number of merged models. The results indicate that as the number of task vectors increases, the percentage of masked weight elements (%) also increases. Notably, this increase is accompanied by consistent improvements in unlearning performance ($D_f$). This suggests that leveraging more task vectors allows for a more accurate identification of the specific weight elements that require modification for effective unlearning. Ultimately, the task vectors generated through our method lead to changes in only 5–10% of the weight elements in the original model via negation. These findings strongly support our argument that not all weight elements are critical for unlearning, and a small subset is sufficient to achieve strong unlearning performance.

Additionally, these experiments also highlight that our technique maintains the performance of the retain set effectively. By selectively targeting only the weight elements essential for unlearning, our method minimizes the impact on the retain set, resulting in better preservation of its performance compared to other merging techniques.

Table C3 shows that larger zero-out values with uniformly merged sparsified task vectors lead to improved unlearning results. TIES-merging and MagMax exhibit fewer zero-out values, and their performance is expected to be outperformed by our method.

Table C1: **Ratio of Zeroed Elements based on the Number of Merged models (Part 1: Cars, DTD, EuroSAT, GTSRB).** The table reports the zero ratio, accuracy of the forget set (Acc $D_f$), and the retain set (Acc $D_r$) across various numbers of task vectors used for merging. Results are presented with averages and standard deviations (std, $\pm$.) from three runs. Results were obtained using ViT-B/32 standard.

| # | Cars | | | DTD | | | EuroSAT | | | GTSRB | | |
|---|---|---|---|---|---|---|---|---|---|---|---|---|
| | % | Acc $D_f(\downarrow)$ | Acc $D_r(\uparrow)$ | % | Acc $D_f(\downarrow)$ | Acc $D_r(\uparrow)$ | % | Acc $D_f(\downarrow)$ | Acc $D_r(\uparrow)$ | % | Acc $D_f(\downarrow)$ | Acc $D_r(\uparrow)$ |
| 30 | 90.34 | 27.40 | 60.40 | 92.94 | 27.18 | 60.45 | 94.05 | 7.85 | 60.22 | 94.76 | 6.20 | 59.98 |
| 25 | $89.68_{\pm 0.04}$ | $26.63_{\pm 0.06}$ | $60.19_{\pm 0.02}$ | $92.21_{\pm 0.10}$ | $27.25_{\pm 0.28}$ | $60.45_{\pm 0.10}$ | $93.37_{\pm 0.09}$ | $8.33_{\pm 0.06}$ | $60.34_{\pm 0.05}$ | $94.13_{\pm 0.02}$ | $6.45_{\pm 0.01}$ | $60.14_{\pm 0.03}$ |
| 20 | $88.80_{\pm 0.21}$ | $26.09_{\pm 0.66}$ | $60.06_{\pm 0.09}$ | $91.45_{\pm 0.13}$ | $27.13_{\pm 0.13}$ | $60.41_{\pm 0.04}$ | $92.65_{\pm 0.05}$ | $8.79_{\pm 0.06}$ | $60.59_{\pm 0.03}$ | $93.32_{\pm 0.03}$ | $6.71_{\pm 0.06}$ | $60.31_{\pm 0.01}$ |
| 15 | $87.48_{\pm 0.17}$ | $26.02_{\pm 0.20}$ | $59.96_{\pm 0.07}$ | $90.15_{\pm 0.12}$ | $27.68_{\pm 0.09}$ | $60.42_{\pm 0.03}$ | $90.84_{\pm 0.28}$ | $8.33_{\pm 0.11}$ | $60.40_{\pm 0.07}$ | $91.93_{\pm 0.09}$ | $6.92_{\pm 0.17}$ | $60.27_{\pm 0.01}$ |
| 10 | $84.91_{\pm 0.51}$ | $26.57_{\pm 0.82}$ | $59.94_{\pm 0.14}$ | $87.52_{\pm 0.31}$ | $27.82_{\pm 0.09}$ | $60.35_{\pm 0.12}$ | $88.25_{\pm 0.51}$ | $8.86_{\pm 0.21}$ | $60.55_{\pm 0.11}$ | $89.24_{\pm 0.19}$ | $8.13_{\pm 0.08}$ | $60.57_{\pm 0.04}$ |
| 5 | $77.12_{\pm 0.30}$ | $30.49_{\pm 0.42}$ | $60.37_{\pm 0.09}$ | $79.86_{\pm 0.60}$ | $28.79_{\pm 0.39}$ | $60.52_{\pm 0.14}$ | $81.51_{\pm 0.26}$ | $9.25_{\pm 0.55}$ | $60.61_{\pm 0.10}$ | $81.86_{\pm 0.32}$ | $8.32_{\pm 0.35}$ | $60.46_{\pm 0.08}$ |

Table C2: **Ratio of Zeroed Elements based on the Number of Merged models (Part 2: MNIST, RESISC45, SUN397, SVHN).** The table reports the zero ratio, accuracy of the forget set (Acc $D_f$), and the retain set (Acc $D_r$) across various numbers of task vectors used for merging. Results are presented with averages and standard deviations (std, $\pm$.) from three runs. Results were obtained using ViT-B/32 standard.

| # | MNIST | | | RESISC45 | | | SUN397 | | | SVHN | | |
|---|---|---|---|---|---|---|---|---|---|---|---|---|
| | % | Acc $D_f(\downarrow)$ | Acc $D_r(\uparrow)$ | % | Acc $D_f(\downarrow)$ | Acc $D_r(\uparrow)$ | % | Acc $D_f(\downarrow)$ | Acc $D_r(\uparrow)$ | % | Acc $D_f(\downarrow)$ | Acc $D_r(\uparrow)$ |
| 30 | 93.96 | 20.50 | 59.90 | 92.86 | 22.61 | 60.47 | 92.20 | 47.19 | 60.56 | 92.40 | 7.18 | 60.86 |
| 25 | $93.30_{\pm 0.10}$ | $20.46_{\pm 0.46}$ | $60.15_{\pm 0.06}$ | $92.27_{\pm 0.04}$ | $23.74_{\pm 0.06}$ | $60.64_{\pm 0.02}$ | $91.68_{\pm 0.09}$ | $46.99_{\pm 0.02}$ | $60.44_{\pm 0.04}$ | $91.70_{\pm 0.02}$ | $7.05_{\pm 0.01}$ | $60.67_{\pm 0.01}$ |
| 20 | $92.49_{\pm 0.08}$ | $19.56_{\pm 0.46}$ | $59.85_{\pm 0.07}$ | $91.50_{\pm 0.06}$ | $22.70_{\pm 0.08}$ | $60.45_{\pm 0.02}$ | $90.89_{\pm 0.12}$ | $47.82_{\pm 0.10}$ | $60.59_{\pm 0.04}$ | $90.68_{\pm 0.04}$ | $6.96_{\pm 0.02}$ | $60.36_{\pm 0.03}$ |
| 15 | $91.02_{\pm 0.06}$ | $20.77_{\pm 0.49}$ | $60.10_{\pm 0.30}$ | $90.13_{\pm 0.09}$ | $23.19_{\pm 0.18}$ | $60.50_{\pm 0.03}$ | $89.60_{\pm 0.33}$ | $47.73_{\pm 0.54}$ | $60.46_{\pm 0.19}$ | $89.20_{\pm 0.27}$ | $7.09_{\pm 0.04}$ | $60.70_{\pm 0.08}$ |
| 10 | $88.22_{\pm 0.47}$ | $19.86_{\pm 0.77}$ | $60.51_{\pm 0.15}$ | $87.56_{\pm 0.28}$ | $22.56_{\pm 0.79}$ | $60.36_{\pm 0.08}$ | $87.23_{\pm 0.31}$ | $48.71_{\pm 0.41}$ | $60.56_{\pm 0.11}$ | $86.03_{\pm 0.20}$ | $7.16_{\pm 0.01}$ | $60.75_{\pm 0.08}$ |
| 5 | $81.35_{\pm 0.25}$ | $22.96_{\pm 0.36}$ | $62.40_{\pm 0.02}$ | $80.20_{\pm 0.52}$ | $24.08_{\pm 1.24}$ | $60.39_{\pm 0.20}$ | $81.50_{\pm 0.10}$ | $49.70_{\pm 0.08}$ | $60.47_{\pm 0.11}$ | $78.93_{\pm 0.18}$ | $7.22_{\pm 0.03}$ | $60.60_{\pm 0.08}$ |

Table C3: **Ratio of Zeroed Elements and the Unlearning Performance.** The table reports the zero ratio and the accuracy of the forget set (Acc $D_f$), comparing these values across several baseline methods. Results were obtained using ViT-B/32 standard.

| Method | Cars | | DTD | | EuroSAT | | GTSRB | | MNIST | | RESISC45 | | SUN397 | | SVHN | |
|---|---|---|---|---|---|---|---|---|---|---|---|---|---|---|---|---|
| | % | $D_f(\downarrow)$ | % | $D_f(\downarrow)$ | % | $D_f(\downarrow)$ | % | $D_f(\downarrow)$ | % | $D_f(\downarrow)$ | % | $D_f(\downarrow)$ | % | $D_f(\downarrow)$ | % | $D_f(\downarrow)$ |
| Task Arithmetic[†] | 47.55 | 29.00 | 47.55 | 30.42 | 47.55 | 10.44 | 47.55 | 9.09 | 47.55 | 21.15 | 47.55 | 30.71 | 47.55 | 50.58 | 47.55 | 7.61 |
| MagMax | 47.55 | 35.59 | 47.55 | 31.86 | 47.55 | 10.51 | 47.55 | 8.42 | 47.55 | 20.14 | 47.55 | 30.69 | 47.55 | 55.36 | 47.55 | 9.33 |
| TIES-Merging | 51.59 | 34.03 | 50.62 | 33.09 | 51.62 | 11.56 | 50.72 | 10.21 | 51.94 | 26.09 | 51.00 | 33.41 | 50.58 | 53.80 | 52.06 | 7.48 |
| **NegMerge (Ours)** | 90.34 | 27.40 | 92.94 | 27.18 | 94.05 | 7.85 | 94.76 | 6.20 | 93.96 | 20.50 | 92.86 | 22.61 | 92.20 | 47.19 | 92.40 | 7.18 |

# D  DIVERSE MODEL POOL

## D.1  ROBUSTNESS ON LARGER MODEL POOL

In the CLIP Unlearning Scenario, we conducted additional experiments to analyze the impact of varying hyperparameters, such as weight decay, learning rates, and label smoothing, as shown in Table D1. Also, in Table D2 we presents the effects of modifying training hyperparameters and RandAugment configurations. Furthermore, we evaluated in seven different model pools, which are detailed in Table D3. In the Standard Classifier Unlearning Scenario, we enhanced the diversity of the fine-tuned models by adjusting RandAugment configurations, as summarized in Table D4.

Table D1: **Learning Rate, Weight Decay, and Label Smoothing Configuration Pool on ViT-B/32 standard in CLIP Unlearning Scenario.** The results were obtained by evaluating 16 models created from a pool of configurations using the following hyperparameter settings: learning rates of 0.0001, 0.00005, 0.00001, and 0.000005, weight decay values of 0.01 and 0.1, and label smoothing to 0 and 0.1.

| Method | Cars | | DTD | | SUN397 | |
|---|---|---|---|---|---|---|
| | Acc $D_f(\downarrow)$ | Acc $D_r(\uparrow)$ | Acc $D_f(\downarrow)$ | Acc $D_r(\uparrow)$ | Acc $D_f(\downarrow)$ | Acc $D_r(\uparrow)$ |
| Task Arithmetic[†] | 33.52 | 60.29 | 29.14 | 60.38 | 51.36 | 60.55 |
| **NegMerge (Ours)** | 30.33 | 60.16 | 26.43 | 59.95 | 47.94 | 60.33 |

Table D2: **RandAugment, Learning Rate, and Weight Decay Configuration Pool on ViT-B/32 standard in CLIP Unlearning Scenario.** The results were obtained by evaluating 8 models created from a pool of configurations using the following hyperparameter settings: RandAugment with $n = 1, 2$ and $m = 1, 5, 10$, learning rates of 0.00001, 0.000005, and 0.000001, and weight decay values of 0.01 and 0.1.

| Method | Cars | | DTD | | EuroSAT | | GTSRB | | MNIST | | RESISC45 | | SUN397 | | SVHN | |
|---|---|---|---|---|---|---|---|---|---|---|---|---|---|---|---|---|
| | $D_f(\downarrow)$ | $D_r(\uparrow)$ | $D_f(\downarrow)$ | $D_r(\uparrow)$ | $D_f(\downarrow)$ | $D_r(\uparrow)$ | $D_f(\downarrow)$ | $D_r(\uparrow)$ | $D_f(\downarrow)$ | $D_r(\uparrow)$ | $D_f(\downarrow)$ | $D_r(\uparrow)$ | $D_f(\downarrow)$ | $D_r(\uparrow)$ | $D_f(\downarrow)$ | $D_r(\uparrow)$ |
| Task Arithmetic[†] | 28.62 | 60.17 | 28.03 | 60.15 | 7.66 | 60.46 | 5.31 | 60.22 | 14.56 | 60.55 | 27.19 | 60.72 | 51.38 | 60.32 | 6.75 | 61.30 |
| **NegMerge (Ours)** | 27.42 | 60.03 | 26.80 | 60.08 | 7.03 | 60.39 | 4.82 | 59.50 | 12.89 | 59.94 | 18.23 | 59.81 | 48.73 | 60.38 | 6.71 | 60.29 |

Table D3: **Average of Seven Different Model Pools.** The model pool details are as follows: Pool 1: RandAugment configurations (n:1–3, m:1–10) with 30 models. Pool 2: Learning rates (1e-04, 1e-05, 5e-05, 5e-06), weight decay (0.01, 0.1), and label smoothing (0, 0.1) with 16 models. Pool 3: Learning rates (1e-04, 1e-05, 5e-05, 5e-06) and weight decay (0.01, 0.1) with 8 models. Pool 4: Learning rates (1e-05, 5e-06, 5e-05) and weight decay (0.01, 0.1) with 6 models. Pool 5: RandAugment (n:1–2, m:5,10), learning rates (1e-04, 1e-05, 5e-05, 5e-06), weight decay (0.01, 0.1), and label smoothing (0, 0.1) with 64 models. Pool 6: RandAugment (n:1–2, m:5,10), learning rates (1e-04, 1e-05, 5e-05, 5e-06), and weight decay (0.01, 0.1) with 32 models. Pool 7: RandAugment (n:1–2, m:1,5,10), learning rates (1e-05, 5e-06, 1e-06), and weight decay (0.01, 0.1) with 8 models.

| Pool | Pool 1 ($D_f \downarrow$) | Pool 2 ($D_f \downarrow$) | Pool 3 ($D_f \downarrow$) | Pool 4 ($D_f \downarrow$) | Pool 5 ($D_f \downarrow$) | Pool 6 ($D_f \downarrow$) | Pool 7 ($D_f \downarrow$) |
|---|---|---|---|---|---|---|---|
| Task Arithmetic[†] | 23.63 | 22.80 | 23.21 | 23.21 | 24.05 | 21.31 | 21.19 |
| **NegMerge (Ours)** | 20.76 | 21.69 | 21.56 | 22.23 | 22.65 | 19.37 | 19.08 |

As shown in Tables D1–D4, unlearning performance improves further compared to the baseline, as expected, with diverse types of hyperparameters. These results highlight that, while the degree of improvement may vary depending on the model pool, using multiple models consistently provides more stable performance gains compared to using a single model.

## D.2  STRATEGY TO CREATE VARIANTS

NegMerge relies on the knowledge encoded in each task vector. Based on our observations, poorly constructed task vectors—such as those trained with unreasonable weight decay—can result in unre-

Table D4: **RandAugment Configuration Pool on ResNet-18 in Standard Classifier Unlearning Scenario.**
The results were obtained by evaluating 5 models created from a pool of configurations using the following
hyperparameter settings: RandAugment with $n = 1$ and $m = 1, 2, 3, 4, 5$.

| Methods | Acc $D_r(\simeq)$ | Acc $D_f(\simeq)$ | Acc $D_{test}(\simeq)$ | MIA($\simeq$) | Avg. Gap($\downarrow$) |
|---|---|---|---|---|---|
| **Retrain *** | 100.00 | 94.76 | 94.26 | 12.88 | 0.00 |
| Task Arithmetic[†] | 97.79 | 95.88 | 91.31 | 9.58 | 2.40 |
| **NegMerge (Ours)** | 97.81 | 95.76 | 91.03 | 10.76 | 2.14 |

liable knowledge and introduce noise into the process. To mitigate this, we recommend constructing
a task vector pool using reasonable and functional hyperparameters, ensuring the vectors are reliable
and contribute effectively to unlearning.

Furthermore, our method inherently produces a model with a well-optimized retain loss. This aligns
with one of our core assumptions discussed in Section 4.3: the merged model should preserve
performance on the retain set. Practitioners can leverage this property by using the retain loss as
a guiding signal during the model generation phase, enabling more effective model merging through
better monitoring and optimization of retain set performance. While we have not yet explored this
idea, we view it as an exciting direction for future research.

## E    MEMORY AND COMPUTATIONAL EFFICIENCY

We intend to address the computational cost from four perspectives: storage, runtime memory,
merge time and inference time complexity. It is important to note that the merging methods we
compare all share the same model pool, so there is no inherent computational overhead from using
multiple models.

### E.1    STORAGE

Although storing all models before merging would be costly in terms of memory, our method does
not require keeping all fine-tuned models stored. During the process of training multiple models,
we only need to check the sign consensus of each element and can store them by adding them to
the existing model. This approach allows our technique to utilize the same amount of storage as the
conventional single-model baseline while performing more effective unlearning. This is not feasible
with other merging methods like TIES-Merging, highlighting the superiority of our approach.

### E.2    RUNTIME MEMORY

Our method offers a distinct advantage in runtime memory usage. As shown in Table C1 and C2,
during the merging process, a significant proportion of the weights in the task vector are zeroed
out, leaving only 5–10% of the total weight elements active. This substantial reduction in active
weights enables the adoption of lightweight storage techniques, such as weight lookup tables, to
further minimize runtime memory requirements. By storing only the active weights, our method
achieves greater efficiency compared to baseline approaches.

### E.3    MERGE TIME

Our method requires checking the sign of each weight element, which takes longer than methods
like MagMax, which only detects maximum values, or Uniform merge, which calculates averages.
However, our method is significantly faster than those requiring more complex operations, such as
TIES-Merging or Greedy Merge. Detailed estimation results can be found in Table 1.

### E.4    INFERENCE TIME COMPLEXITY

Achieving optimal performance in Task Arithmetic often requires multiple training rounds for hy-
perparameter validation due to hyperparameter sensitivity shown in Figure 1. This sensitivity often
necessitates $n$ iterations of hyperparameter tuning during the validation process. Additionally, de-
termining the scaling coefficient—an inference hyperparameter—requires 20 inferences per task

vector. With $n$ validation rounds, the total inference cost scales to $20 \times n$, resulting in a complexity of $O(mn)$, where $m$ represents the number of inferences per round.

In contrast, our approach eliminates the need for repeated inferences per task vector. It requires only 20 inferences to determine the scaling coefficient, regardless of the number of task vectors or validation rounds. This fixed inference cost reduces the complexity to $O(m)$, making our method significantly faster and more efficient in hyperparameter tuning scenarios.

To better emulate a realistic validation process, we conducted experiments by varying $n$ values from 5 to 30. Tables C1 and C2 demonstrates that our method remains robust across diverse $n$ settings.

# F  THEORY

## F.1  THEORETICAL CONJECTURE

Let $\theta_{ori}$ and $\theta_{ft}$ denote the weights of the pre-trained model and a fine-tuned model, respectively. We have the formulation $\theta_{unlearn} = \theta_{ori} - \lambda \tau_{\text{merged}}$, where $\tau_{\text{merged}}$ is from Eq. 1 - our consensually merged task vector. We argue that achieving larger zero-out values with sparsified consensus editing signals in $\tau_{\text{merged}}$ could lead to unlearning performance improvements, based on the following fundamental claims: (1) Weight-wise unanimous consensus merging reduces non-zero values and gives a robust $\tau_{\text{merged}}$; the larger zero-out values in $\tau_{\text{merged}}$ contribute to stable merging. (2) There exists a stable merged point $\theta_{unlearn}^*$, which enjoys better unlearning results as $k$ as the number of task vector $\tau_k$ increases.

## F.2  INFORMAL PROOF

As the number of task vectors $\tau_k$ in $\tau_{\text{merged}}$ increases, the non-zero values decrease because our consensus operation performs like an AND operation. The robustness of $\tau_{\text{merged}}$ increases upon merging, as sparse weights are merged uniformly, which enjoys inherently more robustness than an individual weight as revealed in (Wortsman et al., 2022; Jang et al., 2025), where the first term of $\tau_{\text{merged}} = 1/n * \sum_k \theta_{ft} - \theta_{ori}$ conducts uniform merge. $\theta_{unlearn}$ moves closer to $\theta_{ori}$, which likely stays lower loss regions due to weights generally holding linear mode connectivity (LMC) (Frankle et al., 2020; Juneja et al., 2022; Entezari et al., 2021). It also mitigates issues that cause fluctuating high losses (i.e., loss barriers) on certain loss surfaces, when weights deviate significantly from $\theta_{ori}$. Therefore, task negation at an improved $\theta_{unlearn}^*$ would likely reside in lower loss regions which leads to yielding a better result. While we do not specify the exact $\theta_{unlearn}^*$, increasing the number of merged task vectors allows the process to approach a closer-to-optimal point as more sparsified merged weights are merged uniformly (Jang et al., 2025).

## F.3  EMPIRICAL BACKUP

Our empirical evidence, shown in Tables C1 and C2, demonstrates that larger zero-out values from uniformly merged sparsified task vectors lead to improved unlearning results. Additionally, as indicated in Table C3, TIES-merging and MagMax exhibit fewer zero-out values, which explains their lower performance compared to our method.

