# OpenReview forum: "NegMerge: Consensual Weight Negation for Strong Machine Unlearning"
_ICLR.cc/2025/Conference — Submitted to ICLR 2025_

### Official Review · Reviewer_6Rn3 · 2024-10-31

**Soundness:** 4
**Presentation:** 4
**Contribution:** 3
**Rating:** 6
**Confidence:** 4

**Summary:**

The article introduces a model aggregation method for unlearning, which relies on task arithmetic. It uses multiple sets of hyperparameters to generate several task vectors for a single forget task, then merges them. During this merge process, parameters with sign conflicts are discarded, as they show low correlation with the forget set. Experiments demonstrate that this method is more effective in unlearning compared to other model merge techniques.

**Strengths:**

*   The paper proposes aggregating multiple models in unlearning, which is more advantageous than using a single model.
*   The method is simple and effective.

**Weaknesses:**

*   The paper lacks an explanation of why the proposed method specifically targets unlearning. From the description, it seems more like an optimization method for task vectors.

*   Naturally, this raises further questions about how this model merge method performs on other tasks, such as multi-task adding and task analogies in task arithmetic.

*   The comparison of computational complexity is puzzling. The paper discusses the complexity of the validation process, but the training process's complexity is of greater concern. Moreover, for vanilla task arithmetic, multiple models are not used, so the (O(mn)) notation is misleading.

**Questions:**

*   The paper uses an averaging method to obtain the final task vector. Has there been any exploration of other methods, such as taking the maximum (similar to MagMax) or minimum values?
*   Why was RandAugment used for the CLIP model instead of varying hyperparameters?

---

> ### Author Response · Authors · 2024-11-25
> **To Reviewer 6Rn3 (1/4)**
>
> We appreciate the recognition that using multiple models is superior to using a single model, and we are also grateful for the mention of our method being both simple and effective.  Below, we will address the reviewer's concerns. We incorporated the following responses into the revised version of the paper.

---

> ### Author Response · Authors · 2024-11-25
> **To Reviewer 6Rn3 (2/4)**
>
> **Q1) Why is the proposed method presented as targeting unlearning when its description suggests it primarily optimizes task vectors?**
>
> **A1.** The primary motivation for this research arose from an observation that existing model merging techniques, such as Greedy Merge, often degrade unlearning performance compared to the baseline. While these techniques generally improve performance in conventional machine learning settings, they lead to performance degradation in machine unlearning scenarios. This discrepancy suggests that task vector addition and negation may require fine-tuned models with fundamentally different specifications. Interestingly, while some of the prior research has explored how to construct fine-tuned models for addition [A, B], relatively few studies have addressed how to create models specifically suited for negation. This gap inspired us to focus on developing fine-tuned models tailored for task vector negation.
>
> To address this, our research explicitly defines the characteristics of fine-tuned models essential for effective task vector negation, filling a critical gap in the existing literature. We found that effective unlearning requires fine-tuned models that fit the forget set accurately while preserving knowledge of the retain set. Furthermore, we showed that such models can be constructed by leveraging sign consistency when merging multiple fine-tuned models. These findings naturally shaped the paper’s focus on task vector negation, or machine unlearning, as its core contribution.
>
> We appreciate the reviewer’s concern about exploring fine-tuned models for both addition and negation, but we believe this topic lies outside the scope of this paper. However, we agree that exploring the construction of fine-tuned models suitable for both addition and negation is an interesting and valuable research direction. We plan to address this topic as part of our future work.
>
> We are grateful for the reviewer’s thoughtful feedback and hope this response addresses the concern. If further clarification is needed, please do not hesitate to let us know.
>
> #### **References**
> - [A] *Adamerging: Adaptive model merging for multi-task learning*, ICLR 2024
> - [B] *Ties-merging: Resolving interference when merging models*, NeurIPS 2023
>
> ---
>
> **Q2) How does this model merging method perform on other tasks, such as multi-task addition and task analogies in task arithmetic?**
>
> **A2.** Table 3 presents our results from the *learning via addition* experiments when the scaling coefficient is fixed to one. We evaluated performance on both the forget and retain sets when task vectors derived from various merging methods are added to the original model.
>
> Our method not only effectively learns new knowledge for the forget set but also maintains the performance of the retain set. Moreover, our technique outperformed existing merging methods on the forget set. This demonstrates that our approach is successful in both forgetting via negation and learning via addition. We believe this effectiveness may be due to the retain set sharing many features with the forget set. Delving deeper into this analysis could be an interesting direction for future research.
>
> Thank you for the constructive question.

---

> ### Author Response · Authors · 2024-11-25
> **To Reviewer 6Rn3 (3/4)**
>
> **Q3) Why does the paper emphasize the computational complexity of the validation process instead of the training process? Additionally, why is the \(O(mn)\) time complexity considered, even though vanilla task arithmetic does not involve multiple models?**
>
> **A3.** The original Task Arithmetic approach requires fine-tuning multiple models during the training process, leading to a training cost comparable to that of our method. Specifically, our experimental results in Figure 1 indicate that machine unlearning using Task Arithmetic is sensitive to hyperparameter configurations. This sensitivity often necessitates \(n\) iterations of hyperparameter tuning during the validation process. Such an assumption aligns with the widely accepted notion that ML algorithms inherently involve training multiple models to fine-tune hyperparameters during validation. Our study proposes a strategy that effectively leverages the models developed during this validation process to improve unlearning. Thus, our analysis emphasizes the efficiency of the validation process.
>
> From this perspective, the computational complexity of Task Arithmetic can indeed be represented as \(O(mn)\). Assuming \(n\) models are derived during the validation process, the original paper (Ilharco et al., 2022) reports that determining the scaling factor for task vectors requires 20 inference steps, leading to a total of \(20 \times n\) inferences. Representing the number of inference steps as \(m\), the computational complexity becomes \(O(mn)\). In contrast, our method requires only 20 inferences regardless of the number of models, resulting in a complexity of \(O(m)\), making it significantly more efficient.
>
> To better emulate a realistic validation process, we conducted experiments by varying \(n\) values from 5 to 30. These experiments demonstrated that our method remains robust across diverse \(n\) settings. For further details, please refer to our response to Reviewer AEkn’s Q3.
>
> We acknowledge that our discussion on this topic was not sufficiently detailed and appreciate the reviewer for pointing this out. In the revised manuscript, we have expanded the explanation to clarify these points further (Appendix E.4). Thank you for your constructive feedback.
>
> ---
>
> **Q4) The paper adopts an averaging method to derive the final task vector. Has there been any investigation into alternative approaches, such as selecting maximum (similar to MagMax) or minimum values?**
>
> **A4.** Thank you for raising this question. We hadn't initially considered using alternative approaches such as max or min instead of avg, so we appreciate the suggestion.
>
> | Method            | Cars | DTD  | EuroSAT | GTSRB | MNIST | RESISC45 | SUN397 | SVHN | Average |
> |--------------------|------|------|---------|-------|-------|----------|--------|------|---------|
> | Best              | 29.0 | 30.4 | 10.4    | 9.1   | 21.2  | 30.7     | 50.6   | 7.6  | 23.6    |
> | NegMerge (min)    | 26.5 | 27.9 | 10.3    | 8.3   | 25.2  | 20.1     | 46.8   | 8.2  | 21.6    |
> | NegMerge (max)    | 28.2 | 27.4 | 10.7    | 7.5   | 28.7  | 26.0     | 48.3   | 7.3  | 23.0    |
> | NegMerge (avg)    | 27.4 | 27.2 | 7.9     | 6.2   | 20.5  | 22.6     | 47.2   | 7.2  | 20.8    |
>
> The table above reports only the performance on the forget set (Df↓). The full table, including both performances on the retain set (Dr↑) and the forget set (Df↓), can be found in the manuscript.
>
> After conducting experiments with ViT-B/32 and standard task arithmetic, we found that our originally proposed averaging method performed the best. This is likely because averaging helps smooth out potential outliers in individual models during merging, resulting in a more stable and effective task vector.
>
> We believe that the choice of how to combine values during merging, whether it’s avg, max, min, or another method, is an important factor in improving performance. Investigating alternative merging strategies further could be an intriguing direction for future work. We have included this experiment in Appendix A.2. Thank you again for your insightful suggestion.

---

> ### Author Response · Authors · 2024-11-25
> **To Reviewer 6Rn3 (4/4)**
>
> **Q5) Why was RandAugment used for the CLIP model instead of varying hyperparameters?**
>
> **A5.** Thank you for your thoughtful question. The primary objective of our study was to demonstrate the effectiveness of model merging in enhancing machine unlearning. To the best of our knowledge, no prior research has shown that model merging can improve unlearning performance. As such, we focused on validating this potential rather than identifying the optimal strategies for constructing effective model pools.
>
> Aligned with this goal, we adopted straightforward approaches depending on the type of experiment. For CLIP-based experiments, we adjusted data augmentation intensities, while for general image classification experiments, we fine-tuned basic hyperparameters such as weight decay, learning rate, and label smoothing. We believed that demonstrating the effectiveness of such simple methods could establish a baseline for the potential improvements in unlearning performance.
>
> While exploring more advanced strategies for constructing model pools is undoubtedly an important research direction, we intentionally left this as a subject for future work at the time of submission. That said, the reviewer’s question prompted us to consider the impact of different model pools on unlearning performance, which we realized could significantly enhance our paper.
>
> To address this, we conducted additional experiments:
> 1. *CLIP fine-tuning*: We varied hyperparameters (e.g., RandAugment, weight decay, learning rates, and label smoothing) and generated results using seven different model pools. ViT-B/32 is used.
> 2. *Standard image classification*: We adjusted RandAugment parameters to explore various model pools. ResNet18 and CIFAR-10 are used.
>
> ---
>
> **CLIP Experiments - Detailed Results Reported**
>
> | Method          | Cars (Df↓)  | DTD (Df↓)  | EuroSAT (Df↓) | GTSRB (Df↓)  | MNIST (Df↓)  | RESISC45 (Df↓) | SUN397 (Df↓)  | SVHN (Df↓) |
> |-----------------|-------------|------------|---------------|-------------|-------------|----------------|--------------|------------|
> | Single Best Model | 28.62      | 28.03      | 7.66          | 5.31        | 14.56       | 27.19          | 51.38        | 6.75       |
> | NegMerge        | 27.42      | 26.80      | 7.03          | 4.82        | 12.89       | 18.23          | 48.73        | 6.71       |
>
> - *Model pool configuration*: randaug (n:1–2, m:1–10), lr (1e-05, 5e-06, 1e-06), wd (0.01, 0.1) – 8 models.
>
> | Method          | Cars (Df↓)  | DTD (Df↓)  | SUN397 (Df↓) |
> |-----------------|-------------|------------|--------------|
> | Single Best Model | 33.52      | 29.14      | 51.36        |
> | NegMerge        | 30.33      | 26.43      | 47.94        |
>
> - *Model pool configuration*: lr (1e-04, 1e-05, 5e-05, 5e-06), wd (0.01, 0.1), ls (0, 0.1) – 16 models.
>
> **CLIP Experiments - Seven Different Model Pools (Avg Results Reported)**
>
> | Pool         | Pool 1 (Df↓) | Pool 2 (Df↓) | Pool 3 (Df↓) | Pool 4 (Df↓) | Pool 5 (Df↓) | Pool 6 (Df↓) | Pool 7 (Df↓) |
> |--------------|--------------|--------------|--------------|--------------|--------------|--------------|--------------|
> | Best         | 23.63        | 22.80        | 23.21        | 23.21        | 24.05        | 21.31        | 21.19        |
> | NegMerge     | 20.76        | 21.69        | 21.56        | 22.23        | 22.65        | 19.37        | 19.08        |
>
> - *Model Pool Configurations*:
>   - Pool 1: randaug (n:1–3, m:1–10) – 30 models.
>   - Pool 2: lr (1e-04, 1e-05, 5e-05, 5e-06), wd (0.01, 0.1), ls (0, 0.1) – 16 models.
>   - Pool 3: lr (1e-04, 1e-05, 5e-05, 5e-06), wd (0.01, 0.1) – 8 models.
>   - Pool 4: lr (1e-05, 5e-06, 5e-05), wd (0.01, 0.1) – 6 models.
>   - Pool 5: randaug (n:1–2, m:5,10), lr (1e-04, 1e-05, 5e-05, 5e-06), wd (0.01, 0.1), ls (0, 0.1) – 64 models.
>   - Pool 6: randaug (n:1–2, m:5,10), lr (1e-04, 1e-05, 5e-05, 5e-06), wd (0.01, 0.1) – 32 models.
>   - Pool 7: randaug (n:1–2, m:1,5,10), lr (1e-05, 5e-06, 1e-06), wd (0.01, 0.1) – 8 models.
>
> **Standard Classification Experiments**
>
> | Method                | Df | Dtest | MIA  | Avg. Gap |
> |-----------------------|--------|----------|------|----------|
> | Retrain               | 94.76  | 94.26    | 12.88| 75.48    |
> | Single Best Model     | 95.88  | 91.31    | 9.58 | 2.40     |
> | NegMerge     | 95.76  | 91.03    | 10.76| 2.14     |
>
> - *Model pool configuration*: randaug (n:1, m:1–5) – 5 models.
>
>
>
> ---
>
> As shown in the tables, unlearning performance consistently improves compared to the baseline methods. These results highlight that while the degree of improvement may vary depending on the model pool, using multiple models consistently provides more stable performance gains than relying on a single model.
>
> We have added the above results in Appendix D. We believe that exploring optimal model pool construction for more effective unlearning is a promising direction for future research. Thank you again for raising this insightful question.

---

> > ### Comment · Reviewer_6Rn3 · 2024-11-27
> >
> > Thank you to the authors for their response. I believe the soundness and presentation of the paper have improved, and therefore, I have increased the corresponding scores.
> >
> > Additionally, I would like to discuss further Q1-A1: The authors have explained that this method is applicable for merging multiple fine-tuned models on the same dataset, and I acknowledge that merging multiple fine-tuned models can be meaningful for negation. However, I am still interested in whether this merging method can combine different task vectors obtained from different datasets. Even if the results are not satisfactory, it would still be valuable as it helps in the comparative analysis of the method's mechanism.

---

> ### Author Response · Authors · 2024-11-29
> **Appreciation for Your Feedback and Further Experiment**
>
> Dear Reviewer 6Rn3,
>
> Thank you for your follow-up comments and for increasing the scores. We greatly appreciate your continued engagement with our work.
> Regarding your question about whether the proposed merging method can combine task vectors obtained from different datasets, we conducted additional experiments to investigate this scenario. The results, presented below, generally show low performance when merging task vectors across different datasets. These results were obtained using the ViT-B-32 architecture and vanilla task arithmetic.
>
> Since we cannot determine which model would perform best in the this setting, we first conducted experiments using the single best model that showed the best negation performance. Additionally, we performed experiments by randomly selecting one model for each dataset.
>
> **Forget 8 Datasets: Best Single Model**
>
> This experiment evaluates merging task vectors from different datasets using the best single model.
>
> | Method      | Cars (Df↓) | DTD (Df↓) | EuroSAT (Df↓) | GTSRB (Df↓) | MNIST (Df↓) | RESISC45 (Df↓) | SUN397 (Df↓) | SVHN (Df↓) |
> |-------------|------------|-----------|---------------|-------------|-------------|----------------|-------------|------------|
> | Zero-shot   | 59.6       | 43.9      | 45.1          | 33.0        | 48.2        | 61.6           | 63.3        | 29.2       |
> | Uniform     | 53.5       | 39.3      | 35.3          | 22.2        | 44.5        | 53.1           | 60.4        | 13.4       |
> | Ties        | 51.6       | 39.3      | 30.6          | 18.7        | 39.4        | 50.3           | 58.9        | 9.7        |
> | MagMax      | 54.0       | 40.9      | 43.2          | 19.6        | 38.2        | 54.9           | 60.1        | 12.2       |
> | NegMerge    | 53.4       | 40.1      | 30.1          | 24.1        | 49.9        | 54.7           | 61.1        | 17.6       |
>
> **Forget 8 Datasets: Random Model**
>
> Here, task vectors from different datasets are combined using a randomly selected model for each dataset. The experiments were repeated ten times per dataset, with averages and standard deviations reported. In each iteration, the randomly selected models differ, introducing variability that is reflected in the reported standard deviations.
>
>
> | Method      | Cars (std) | DTD (std) | EuroSAT (std) | GTSRB (std) | MNIST (std) | RESISC45 (std) | SUN397 (std) | SVHN (std) |
> |-------------|------------------|----------------|---------------------|------------------|------------------|---------------------|------------------|-----------------|
> | Zero-shot   | 59.6 (–)         | 43.9 (–)       | 45.1 (–)            | 33.0 (–)         | 48.2 (–)         | 61.6 (–)            | 63.3 (–)         | 29.2 (–)        |
> | Uniform     | 51.90 (0.72)     | 38.93 (0.37)   | 37.57 (1.65)        | 19.71 (0.77)     | 43.60 (1.88)     | 51.16 (0.71)        | 59.98 (0.22)     | 13.39 (1.65)    |
> | Ties        | 53.14 (0.66)     | 40.04 (0.42)   | 33.75 (1.87)        | 21.10 (1.26)     | 39.74 (2.34)     | 52.00 (0.96)        | 60.11 (0.39)     | 13.64 (1.96)    |
> | MagMax      | 53.53 (0.16)     | 40.57 (0.24)   | 43.73 (1.23)        | 20.01 (0.83)     | 38.04 (1.51)     | 54.74 (0.22)        | 60.30 (0.08)     | 13.99 (1.80)    |
> | NegMerge    | 53.56 (0.37)     | 40.01 (0.47)   | 32.34 (1.05)        | 22.89 (1.01)     | 46.44 (1.68)     | 54.98 (0.47)        | 61.12 (0.17)     | 17.81 (1.18)    |
>
>
> These findings illustrate that all methods, including ours and others, face challenges in this scenario, as they are not designed with this kind of unlearning setting in mind. Despite these difficulties, our method does not exhibit greater struggles compared to other approaches.
>
> Thank you once again for your valuable insights and constructive feedback.
>
>
> Best regards,
>
> NegMerge Authors

---

### Official Review · Reviewer_EBwf · 2024-11-01

**Soundness:** 2
**Presentation:** 2
**Contribution:** 2
**Rating:** 6
**Confidence:** 4

**Summary:**

This paper proposes a method for machine unlearning in deep learning models. Unlike existing methods that rely on a single fine-tuned model's task vector, NegMerge combines task vectors from multiple fine-tuned models trained with different hyperparameters, keeping only elements with consistent signs. This approach enables robust forgetting of the forget set while minimizing the impact on the retain set.

**Strengths:**

* Easy to follow
* Simple yet effective performance : intuitive approach
* Experiments with different archihtectures (e.g CLIP and ResNet) which are commonly used.

**Weaknesses:**

* Lack of technical contribution
: They introduce practical approach but still lacks the theoretical depth. The paper may need for more analysis why sign-consistency aligns with forget set or exploring alternative merging method.
* Reliance on empirical evidence
: As mentioned in Section 4.3, the paper relies on empirical results regarding performance on the forget set and retain set. The experimental results show that while the method achieves effective unlearning on the forget set, it compromises performance on the retain set, showing degradation in preserving non-targeted information compared to baselines. Without further analysis, it remains unclear whether these findings generalize across diverse datasets and model architectures.

**Questions:**

* As mentioned in the paper, only the final layer of CLIP's text encoder remains frozen during fine-tuning. Given that the CLIP model’s image-text alignment might still influence the unlearning process, I wonder if unfreezing the final layer would provide any additional benefits. Was this approach tested, or was it determined that freezing the final layer would not significantly impact the results? If so, what was the reasoning behind this choice?

* The method relies on multiple fine-tuned models, which can increase memory costs due to the need of several sets of weights. Compared to baselines, could you compare how computationally efficient this approach is and whether this efficiency is associated with performance improvements across the models?

---

> ### Author Response · Authors · 2024-11-25
> **To Reviewer EBwf (1/4)**
>
> We are very pleased that you found our paper to be clearly written. Additionally, we appreciate your recognition of our method as being simple yet effective and intuitive. We are also grateful that you highlighted the practicality and versatility of our approach through experiments across various architectures, which underscores its applicability.
>
> Below, we address your specific concerns in detail. All the following responses have been incorporated into the revised version of the paper.

---

> ### Author Response · Authors · 2024-11-25
> **To Reviewer EBwf (2/4)**
>
> **Q1) Could the paper enhance its analysis of why sign-consistency aligns with the forget set, potentially incorporating a theoretical perspective, or explore alternative merging methods to further strengthen its technical contributions?**
>
> **A1.** Here’s the updated theorem with the definition of the merged weights included:
>
>  **Our conjecture**
>
> Let $\theta\_{ori}$ and $\theta^k\_{ft}$ denote the weights of the pre-trained model and a fine-tuned model, respectively. We have the formulation $\theta\_{unlearn} = \theta\_{ori} - \lambda \tau\_{merged}$, where $\tau\_{merged}$ is from Eq.(1) - our consensually merged task vector. We argue that achieving larger zero-out values with sparsified consensus editing signals in $\tau\_{merged}$ could lead to unlearning performance improvements, based on the following fundamental claims:
> - Weight-wise unanimous consensus merging reduces non-zero values and gives a robust $\tau\_{merged}$; the larger zero-out values in $\tau\_{merged}$ contribute to stable merging.
> - There exists a stable merged point $\theta^*\_{unlearn}$, which enjoys better unlearning results as $k$ as the number of task vectors $\tau\_k$ increases.
>
> **Informal theoretical proof**
>
> As the number of task vectors $\tau\_k$ in $\tau\_{merged}$ increases, the non-zero values decrease because our consensus operation performs like an AND operation. The robustness of $\tau\_{merged}$ increases upon merging, as sparse weights are merged uniformly, which enjoys inherently more robustness than an individual weight as revealed in [D, E], where the first term of $\tau\_{merged}=1/n * \sum\_k \theta^k\_{ft} - \theta\_{ori}$ conducts uniform merge.
> $\theta\_{unlearn}$ moves closer to $\theta\_{ori}$, which likely stays lower loss regions due to weights generally holding linear mode connectivity (LMC) [A, B, C]. It also mitigates issues that cause fluctuating high losses (i.e., loss barriers) on certain loss surfaces, when weights deviate significantly from $\theta\_{ori}$. Therefore, task negation at an improved $\theta^*\_{unlearn}$ would likely reside in lower loss regions which leads to yielding a better result. While we do not specify the exact $\theta^*\_{unlearn}$, increasing the number of merged task vectors allows the process to approach a closer-to-optimal point as more sparsified merged weights are merged uniformly [E].
>
> **References**
> - [A] *Linear Mode Connectivity and the Lottery Ticket Hypothesis*, ICML 2020
> - [B] *Linear Connectivity Reveals Generalization Strategies*, ICLR 2023
> - [C] *The Role of Permutation Invariance in Linear Mode Connectivity of Neural Networks*, ICLR 2022
> - [D] *Model soups: averaging weights of multiple fine-tuned models improves accuracy without increasing inference time*, ICML 2022
> - [E] *Model Stock: All we need is just a few fine-tuned models*, ECCV 2024
>
>  **Empirical Backup**
> Our empirical evidence supports the informal claim: larger zero-out values with uniformly merged sparsified task vectors lead to improved unlearning results. The table below presents the averaged results across all datasets, with the full results for each dataset included in the paper (*Tables C1* and *C2*). Starting with a pool of 30 models, we randomly selected \( n \) models three times for each experiment, and the mean and standard deviation for each metric are reported.
>
> The results indicate that as the zero ratio increases, unlearning performance improves (lower $\( D_f \)$). This demonstrates the effectiveness of larger zero-out values in achieving better unlearning outcomes.
>
> | **# Task Vectors** | **Zero Ratio (std)** | **Df↓ (std)** |
> |--------------------|----------------------|---------------|
> | 30 | 92.94         | 20.76         |
> | 25 | 92.29 (0.06)  | 20.86 (0.12)  |
> | 20 | 91.47 (0.09)  | 20.72 (0.20)  |
> | 15 | 90.04 (0.17)  | 20.97 (0.23)  |
> | 10 | 87.37 (0.35)  | 21.21 (0.40)  |
> | 5  | 80.29 (0.32)  | 22.60 (0.43)  |
>
>
> Additionally, we observed that methods like TIES-merging and MagMax exhibit lower zero-out ratios compared to our approach, resulting in diminished unlearning performance. The table below further supports this observation, showing the relationship between the zero-out ratio and $\( D_f \)$. Our method, NegMerge, achieves the highest zero ratio and consistently outperforms other methods in unlearning performance. These results underscore the advantage of higher zero-out ratios achieved by NegMerge, enabling superior unlearning effectiveness compared to alternative methods.
>
> | **Method**       | **Zero Ratio** | **Df↓**   |
> |-------------------|----------------|-----------|
> | Single best  | 47.55    | 23.63 |
> | MagMax       | 47.55    | 25.24 |
> | TIES         | 51.27    | 26.21 |
> | NegMerge     | 92.94    | 20.76 |
>
> We have included this discussion in Appendix F. This significantly enhances the quality of our paper. Thank you for the constructive feedback.

---

> ### Author Response · Authors · 2024-11-25
> **To Reviewer EBwf (3/4)**
>
> **Q2) Does the method's effective unlearning on the forget set inevitably lead to compromised performance on the retain set? How does this degradation compare to baseline methods in preserving non-targeted information?**
>
> **A2.** We would like to clarify that our method does not result in a reduction of accuracy on the retain set compared to existing techniques. Following the established evaluation practices by Ilharco et al. (2022), we ensured a fair comparison. This protocol aims to maintain the performance on the retain set at approximately 95% of the original accuracy, with the assessment focusing on performance on the forget set. Specifically, we tested 20 scaling coefficients on the validation set, selecting the coefficient that most closely approached 95% of the original accuracy. Achieving exactly 95% is challenging, which leads to slight variations among methods. Consequently, all methods show retain set accuracies within the range of 60-61%. For a detailed explanation of this evaluation protocol, please refer to Appendix B.1 of Ilharco et al. (2022).
>
> Given this controlled performance on the retain set, it is appropriate to compare the effectiveness of techniques based solely on their forget set performance. From this perspective, our method has demonstrated superior unlearning performance compared to existing model merging techniques.
>
> We apologize for any confusion caused by the lack of clarity on this point in our paper. We have updated the manuscript to include the above discussion (Section 4.2).
>
> ---
>
> **Q3) Do the experimental findings generalize across a wider range of datasets and model architectures beyond those tested?**
>
> **A3.** To ensure a fair comparison, we followed the experimental settings of existing baseline papers. Specifically, we conducted experiments on eight datasets using the CLIP ViT-B32/16/L-14 architectures as in Ilharco et al., 2022, and on one dataset using the ResNet architecture as in Fan et al., 2024. In summary, our experiments spanned four different architectures across nine datasets. We believe that these experiments effectively demonstrate the generalizability of our technique.
>
> However, upon reviewing the reviewers' feedback, we acknowledge the need for additional experiments on other architectures and datasets. In particular, for the standard classification setting, we only utilized one dataset and one architecture, which may not fully validate the generalizability of our method. To address this, we conducted additional experiments using ResNet-18 on the CUB dataset, which comprises 200 bird species. This dataset is important for evaluating unlearning performance in fine-grained image classification tasks. The results, shown in the table below, indicate that our method outperforms existing techniques. We added this results in Table A8.
>
> | Method                   | Dr    | Df    | Dtest | MIA   | Avg.Gap |
> |--------------------------|-------|-------|-------|-------|---------|
> | Retrain                 | 78.55 | 56.43 | 74.61 | 80.47 | 0.00    |
> | Gradient Ascent         | 66.75 | 57.26 | 66.60 | 67.61 | 8.38    |
> | Boundary Shrink         | 66.88 | 61.60 | 64.14 | 100.00 | 11.71   |
> | Boundary Expanding      | 65.32 | 61.60 | 58.80 | 73.62 | 10.27   |
> | Random Labeling         | 64.13 | 57.43 | 59.54 | 71.79 | 9.79    |
> | SalUn                   | 66.69 | 59.60 | 63.88 | 74.46 | 7.94    |
> | Task Arithmetic (Best)  | 74.68 | 58.60 | 70.56 | 100.00 | 7.41    |
> | Uniform Merge           | 73.94 | 56.93 | 69.78 | 100.00 | 7.37    |
> | NegMerge     | 74.64 | 58.26 | 70.69 | 100.00 | 7.30    |
>
> Additionally, to further verify the generalizability of our approach beyond ResNet-18, we performed experiments on the CIFAR-10 dataset using the VGG-16 architecture. The following table summarizes the preliminary results, which demonstrate the superior performance of our method. The complete results can be found in Table A9 of the manuscript.
>
> | Method                  | Dr    | Df    | Dtest | MIA   | Avg.Gap |
> |-------------------------|-------|-------|-------|-------|---------|
> | Retrain                 | 99.99 | 94.02 | 93.06 | 10.36 | 0.00    |
> | Gradient Ascent         | 99.37 | 99.07 | 93.63 | 1.36  | 3.81    |
> | Boundary Shrink         | 99.40 | 99.20 | 93.68 | 1.38  | 3.84    |
> | Boundary Expanding      | 99.39 | 99.20 | 93.68 | 1.42  | 3.84    |
> | Task Arithmetic (Best)  | 97.26 | 94.90 | 90.10 | 10.34 | 1.64    |
> | NegMerge     | 98.00 | 95.74 | 91.01 | 10.10 | 1.50    |
>
>
>
> Through these additional experiments involving the CUB dataset and VGG-16 architecture in the standard classification setting, we have further validated the generalizability of our proposed method across diverse datasets and model architectures. We believe these results significantly enhance the quality of our work. Thank you for the constructive feedback.

---

> ### Author Response · Authors · 2024-11-25
> **To Reviewer EBwf (4/4)**
>
> **Q4) Was unfreezing the final layer of CLIP's text encoder considered during fine-tuning? If not, what was the reasoning for keeping it frozen?**
>
> **A4.** We follow the training practices suggested by Task Arithmetic (Ilharco et al., 2022), keeping the last layer of the CLIP text encoder frozen. This approach simplifies the training process by avoiding the introduction of additional learnable parameters and preserving the integrity of the pretrained features. Research has also shown that freezing this layer has minimal impact on performance (for details, see Appendix B.1 of Ilharco et al., 2022).
>
> We acknowledge that our explanation was lacking and have included this reasoning in the paper (Section 4.2). We hope this addresses your concerns, and we are open to considering alternative approaches if you have further suggestions. Thank you for your input.
>
> ---
>
> **Q5) How does the method's memory and computational efficiency compare to baselines, given the need for multiple fine-tuned models? Is this efficiency linked to performance improvements?**
>
> **A5.** We intend to address the computational cost from four perspectives: storage, runtime memory, and merge time. It is important to note that the merging methods we compare all share the same model pool, so there is no inherent computational overhead from using multiple models.
>
> **Storage:**
> As the reviewer pointed out, storing all models before merging would indeed be costly in terms of memory. However, our method does not require keeping all fine-tuned models stored. During the process of training multiple models, we only need to check the sign consensus of each element and can store them by adding them to the existing model. This approach allows our technique to utilize the same amount of storage as the conventional single-model baseline while performing more effective unlearning. This is not feasible with other merging methods like TIES-merge, highlighting the superiority of our approach.
>
> **Runtime Memory:**
> Our method offers a distinct advantage in runtime memory usage. During the merging process, a significant proportion of the weights in the task vector are zeroed out, leaving only 5–10% of the total weight elements active (for more details, please refer to our response to Reviewer AEkn's Q3). This substantial reduction in active weights enables the adoption of lightweight storage techniques, such as weight lookup tables, to further minimize runtime memory requirements. By storing only the active weights, our method achieves greater efficiency compared to baseline approaches.
>
> **Merge Time:**
> Our method requires checking the sign of each weight element, which takes longer than methods like MagMax, which only detects maximum values, or Uniform merge, which calculates averages. However, our method is significantly faster than those requiring more complex operations, such as TIES-Merging or Greedy Merge. Detailed estimation results can be found in Table 1.
>
> **Inference Time Complexity:**
> Achieving optimal performance in Task Arithmetic (Ilharco et al., 2022) often requires multiple training rounds for hyperparameter validation. Additionally, determining the scaling coefficient—an inference hyperparameter—requires 20 inferences per task vector. With \(n\) validation rounds, the total inference cost scales to \(20 \times n\), resulting in a complexity of \(O(mn)\), where \(m\) represents the number of inferences per round.
>
> In contrast, our approach eliminates the need for repeated inferences per task vector. It requires only 20 inferences to determine the scaling coefficient, regardless of the number of task vectors or validation rounds. This fixed inference cost reduces the complexity to \(O(m)\), making our method significantly faster and more efficient in hyperparameter tuning scenarios (for more details, please refer to our response to Reviewer 6Rn3’s Q3).
>
> **Is this efficiency linked to performance improvements?:**
> We have observed that the proportion of elements in the task vector that are zeroed out after merging directly affects performance. For further details, please refer to our responses to Reviewer EBwf’s Q1 and to Reviewer AEkn's Q3.
>
> Aside from this, we have not found evidence that computational efficiency is directly linked to performance. That is, increasing computational resources does not necessarily enhance performance, nor does reducing computational resources necessarily degrade it.
>
> On the other hand, when compared to existing baseline methods that select a single best model, using multiple models appears to result in computational overhead that increases linearly with the number of models. However, as mentioned above, our technique does not introduce significant computational overhead because the baseline model's validation process for selecting the single best model also involves fine-tuning multiple models.
>
> We have included this discussion in the final version (Appendix E). Thank you for the constructive question.

---

> > ### Comment · Reviewer_EBwf · 2024-11-25
> >
> > The authors addressed my concerns well in the rebuttal. I will raise my rating.

---

> > > ### Author Response · Authors · 2024-11-25
> > > **Thank You Very Much for Your Feedback**
> > >
> > > Dear Reviewer EBwf,
> > >
> > > Thank you for your prompt response and for the updated evaluation of our paper. We truly appreciate your time and effort in reassessing our work. Your initial feedback was incredibly helpful in improving our paper.
> > >
> > > If there are any further improvements or unresolved issues that could affect the review, we would be grateful to hear your suggestions. We are committed to addressing any remaining concerns.
> > >
> > > Thank you once again for your support.
> > >
> > > Best regards,
> > > The Authors

---

### Official Review · Reviewer_s5f2 · 2024-11-03

**Soundness:** 4
**Presentation:** 4
**Contribution:** 3
**Rating:** 6
**Confidence:** 3

**Summary:**

The paper extends the existing work of machine unlearning by generating a task vector based on a forget set for finetuning and subtracting it from the original model. The new proposal gives an approach to merge finetuned candidates by merging only the components of the task vectors with consistent signs.
One advantage of the approach is computational efficiency in the merging step.
The model both shows improvements on unlearning tasks and maintains performance on the retain set.

**Strengths:**

The paper has sufficient numerical results to support the claim for advantages of the proposal on several tasks, including better unlearning, less degradation on the retain set, and computation efficiency. These experiments show concrete evidence, especially the appendix.
For “relationship with TIES-merging,” it is stated that “elements with inconsistent signs across task vectors are more closely related to the retain set than the forget set.” This is an essential claim for this paper, which is indicated by aggregated metrics of two methods and visualization of model activation during inference.

**Weaknesses:**

The evaluation set of forget and retain do not have clear separability and they may have overlap - ImageNet still requires knowledge relevant to the eight tasks of fine-grained datasets and the CIFAR-10 is sampled randomly. Though the higher accuracy for the current retain set can still be a positive evidence, some eval sets completely irrelevant to the forget set may be more strong.

**Questions:**

The paper shows the feasibility of the theory. My personal intuition is that the most promising application of machine unlearning is safety and privacy. However, the experiments focus on some functional recognition and, thus, have a retain eval set with overlap with the forget set. Is there any reason for not having related experiments?

---

> ### Author Response · Authors · 2024-11-25
> **To Reviewer s5f2 (1/2)**
>
> Thank you for acknowledging the strengths of our approach, including the computational efficiency of the merging step, the robust numerical results supporting our claims, and the concrete evidence provided by our experiments. We value your perspective and address your specific concerns in detail below. All the following responses have been incorporated into the revised version of the paper.

---

> ### Author Response · Authors · 2024-11-25
> **To Reviewer s5f2 (2/2)**
>
> **Q1) My personal intuition is that the most promising application of machine unlearning is safety and privacy. However, the experiments only focus on some functional recognition.**
>
> **A1.** We agree that the most promising applications of machine unlearning lie in safety and privacy. For this reason, we included the Membership Inference Attack (MIA) metric in our evaluation. The MIA metric assesses the model's vulnerability to privacy breaches by measuring whether an attacker can infer if specific data was used during training.
>
> As shown in Table 2, our approach demonstrates strong performance on the MIA metric, highlighting its effectiveness not only in utility but also in preserving privacy. This underscores that our method is designed to address both utility and privacy concerns, reinforcing its broader applicability to machine unlearning tasks.
>
> If there are additional privacy-related evaluations or specific scenarios you would like us to explore, we would be happy to consider them as part of our ongoing analysis. Thank you for drawing attention to this critical dimension.
>
> ---
>
> **Q2) The current experiments only have a retain eval set with overlap with the forget set. Though the higher accuracy for the current retain set can still be a positive evidence, some eval sets completely irrelevant to the forget set may be more strong. Is there any reason for not having related experiments?**
>
> **A2.** The reviewer’s concern is valid. Our experimental setting does include cases where the distribution of images to be forgotten is not significantly different from that of the retain set. For example, there is substantial overlap between the Stanford Cars dataset and ImageNet.
>
> However, we do not believe this poses a problem. Task Arithmetic (Ilharco et al., 2022) addressed this issue by removing many overlapping images in their experiments and reported no significant difference in performance, whether the overlapping images were removed or not. For more details, please see Appendix B.5 of Ilharco et al. (2022).
>
> If you have specific dataset suggestions or scenarios in mind, we would be happy to explore them during the ongoing discussion. Thank you again for highlighting this opportunity for improvement.

---

> > ### Comment · Reviewer_s5f2 · 2024-11-25
> >
> > Thanks for your follow-up. Your responses have made me confident taht this paper is a solid 6 and a potential 7. Yet, considering the theory part, I do not plan to upgrade my rating to 8 (the options of upgrading are limited)

---

> > > ### Author Response · Authors · 2024-11-26
> > > **Gratitude for Your Feedback and Follow-up on Theoretical Explanation**
> > >
> > > Dear Reviewer s5f2,
> > >
> > > Thank you for your thoughtful and constructive feedback, as well as your positive evaluation of our manuscript. We truly value your insights, which have allowed us to reflect deeply on the critical aspects of unlearning from a privacy perspective and the dataset composition. These reflections have significantly enriched our understanding and strengthened the manuscript.
> > >
> > > We recognize your observation regarding potential areas for improvement in the theoretical explanation and are eager to address this. While we have included reasoning to explain why our method works, along with informal proofs and empirical evidence to address concerns raised by other reviewers (e.g., Reviewer AEkn’s Q2 and Reviewer EBwf’s Q1), we acknowledge that certain aspects of the explanation may lack sufficient rigor.
> > >
> > > We would greatly appreciate any additional clarification or specific suggestions you could provide regarding areas within the theoretical explanation that might require further refinement. Your insights would greatly help us ensure the manuscript meets a higher standard of rigor and clarity.
> > >
> > > Once again, thank you for your valuable feedback and for taking the time to share your thoughts. We are sincerely grateful for the opportunity to improve our work with your feedback.
> > >
> > > Best regards,
> > > NegMerge Authors

---

### Official Review · Reviewer_AEkn · 2024-11-08

**Soundness:** 3
**Presentation:** 4
**Contribution:** 3
**Rating:** 6
**Confidence:** 4

**Summary:**

This paper focuses on the machine unlearning problem, which aims to selectively remove knowledge from a pre-trained model without retraining from scratch. Instead of subtracting a single task vector from a pre-trained model, this paper proposed to use multiple task vectors with different hyperparameters. Multiple task vectors are merged by averaging elements with the same signs, while the remaining elements with different signs are replaced with zeros. Experimental results showed the effectiveness of merging multiple models during the unlearning process. More importantly, the elements with the same signs have more impact than the elements with conflict signs.

**Strengths:**

* This paper tackles an important and relatively new topic of machine unlearning. It would be very impactful to remove specific data slices from a pre-trained model, without retraining from scratch due to the training cost.
* This paper proposed an intuitively simple idea to merge multiple fine-tuned task vectors into a single one. Empirical results showed the effectiveness.
* Not only the classification results are reported, but also the Membership Inference Attack (MIA) metric is used to assess privacy protection. This could be more useful in practice than just checking classification results.

**Weaknesses:**

* The core idea of using task vector negation for machine unlearning was proposed in (Ilharco et al., 2022). The idea proposed in this paper is a bit incremental, though coming up with this simple idea could still be non-trivial. It would be more convincing to augment this paper with more in depth discussion and analysis of why the proposed method could work.
* Evaluation of a few design choices are missing, please refer to the following questions section for more details.

**Questions:**

* Could you provide stats about how much ratio of the merged task vector becomes ratio? Especially when tuning the number of task vectors (e.g. 5, 10, 20, 30), would that have a significant impact on the zero ratio of the merged vector?
* How do you decide on the hyperparameters? For example, augmentation variants are used for CLIP finetuning but weight decay / epochs / label smoothing are used for standard image classifications. Is it sensitive to choose the hyperparameters to create a set of task vectors? What're the recommendations for a practitioner to create variants?

---

> ### Author Response · Authors · 2024-11-25
> **To Reviewer AEkn (1/6)**
>
> We deeply appreciate the positive reviews and valuable and insightful comments to encourage our work. We have carefully read the comment and addressed the concern through this response. All the following responses have been incorporated into the revised version of the paper.

---

> ### Author Response · Authors · 2024-11-25
> **To Reviewer AEkn (2/6)**
>
> **Q1) The idea proposed in this paper is a bit incremental, though coming up with this simple idea could still be non-trivial.**
>
> **A1.** Our research aims to emphasize several key messages that have not been addressed in prior studies.
>
> First, we provide guidelines for constructing task vectors for *Negation*. While the idea of *Forgetting via Negation* was introduced in *Task Arithmetic* (Ilharco et al., 2022), there has been no discussion on how to construct task vectors to achieve effective unlearning. We address this gap by offering new insights. As discussed in Section 3.1 (*Motivation*), we identify that for effective forgetting, the fine-tuned model must adhere to the following principle: it should maintain high performance on the forget set while avoiding degradation on the retain set. This claim is validated in Section 4.3 (*Empirical Analysis*). These insights provide a deeper understanding of the challenges associated with removing knowledge of data that needs to be forgotten.
>
> Recognizing these challenges, we propose a novel method for obtaining such a fine-tuned model. Specifically, we hypothesize that each element of the weight vector carries varying levels of information related to the forget set. In other words, some weight elements are more strongly associated with the forget set, while others are less so. Based on this hypothesis, identifying the weight elements strongly tied to the forget set becomes crucial. We demonstrate that utilizing sign consensus across task vectors can effectively achieve this, and through extensive experiments, we show that this approach effectively removes knowledge of the data that needs to be forgotten. While similar frameworks like *TIES* or *MagMax* exist, they have not proven effective for unlearning. As the reviewer noted, our approach is simple but non-trivial.
>
> We hope this clarifies the originality and significance of our contributions. Thanks to the reviewer's thoughtful feedback, we were able to express these points more effectively. Thank you for your careful consideration.

---

> ### Author Response · Authors · 2024-11-25
> **To Reviewer AEkn (3/6)**
>
> **Q2) To strengthen the paper, it would be beneficial to include a more detailed discussion and analysis explaining why the proposed method is effective.**
>
> **A2.** Here’s the updated theorem with the definition of the merged weights included:
> **Our conjecture**
>
>
> Let $\theta\_{ori}$ and $\theta^k\_{ft}$ denote the weights of the pre-trained model and a fine-tuned model, respectively. We have the formulation $\theta\_{unlearn} = \theta\_{ori} - \lambda \tau\_{merged}$, where $\tau\_{merged}$ is from Eq.(1) - our consensually merged task vector. We argue that achieving larger zero-out values with sparsified consensus editing signals in $\tau\_{merged}$ could lead to unlearning performance improvements, based on the following fundamental claims:
> - Weight-wise unanimous consensus merging reduces non-zero values and gives a robust $\tau\_{merged}$; the larger zero-out values in $\tau\_{merged}$ contribute to stable merging.
> - There exists a stable merged point $\theta^*\_{unlearn}$, which enjoys better unlearning results as $k$ as the number of task vectors $\tau\_k$ increases.
>
> **Informal proof**
> As the number of task vectors $\tau\_k$ in $\tau\_{merged}$ increases, the non-zero values decrease because our consensus operation performs like an AND operation. The robustness of $\tau\_{merged}$ increases upon merging, as sparse weights are merged uniformly, which enjoys inherently more robustness than an individual weight as revealed in [D, E], where the first term of $\tau\_{merged}=1/n * \sum\_k \theta^k\_{ft} - \theta\_{ori}$ conducts uniform merge.
> $\theta\_{unlearn}$ moves closer to $\theta\_{ori}$, which likely stays lower loss regions due to weights generally holding linear mode connectivity (LMC) [A, B, C]. It also mitigates issues that cause fluctuating high losses (i.e., loss barriers) on certain loss surfaces, when weights deviate significantly from $\theta\_{ori}$. Therefore, task negation at an improved $\theta^*\_{unlearn}$ would likely reside in lower loss regions which leads to yielding a better result. While we do not specify the exact $\theta^*\_{unlearn}$, increasing the number of merged task vectors allows the process to approach a closer-to-optimal point as more sparsified merged weights are merged uniformly [E].
>
>   - *References*
>     - [A] *Linear Mode Connectivity and the Lottery Ticket Hypothesis*, ICML 2020
>     - [B] *Linear Connectivity Reveals Generalization Strategies*, ICLR 2023
>     - [C] *The Role of Permutation Invariance in Linear Mode Connectivity of Neural Networks*, ICLR 2022
>     - [D] *Model soups: averaging weights of multiple fine-tuned models improves accuracy without increasing inference time*, ICML 2022
>     - [E] *Model Stock: All we need is just a few fine-tuned models*, ECCV 2024
>
> **Empirical Backup**
> Our empirical evidence supports the informal claim: larger zero-out values with uniformly merged sparsified task vectors lead to improved unlearning results. The table below presents the averaged results across all datasets, with the full results for each dataset included in the paper (*Tables C1* and *C2*). Starting with a pool of 30 models, we randomly selected \( n \) models three times for each experiment, and the mean and standard deviation for each metric are reported.
>
> The results indicate that as the zero ratio increases, unlearning performance improves (lower $\( D_f \)$). This demonstrates the effectiveness of larger zero-out values in achieving better unlearning outcomes.
>
> | **# Task Vectors** | **Zero Ratio (std)** | **Df↓ (std)** |
> |--------------------|----------------------|---------------|
> | 30 | 92.94         | 20.76         |
> | 25 | 92.29 (0.06)  | 20.86 (0.12)  |
> | 20 | 91.47 (0.09)  | 20.72 (0.20)  |
> | 15 | 90.04 (0.17)  | 20.97 (0.23)  |
> | 10 | 87.37 (0.35)  | 21.21 (0.40)  |
> | 5  | 80.29 (0.32)  | 22.60 (0.43)  |
>
>
> Additionally, we observed that methods like TIES-merging and MagMax exhibit lower zero-out ratios compared to our approach, resulting in diminished unlearning performance. The table below further supports this observation, showing the relationship between the zero-out ratio and $\( D_f \)$. Our method, NegMerge, achieves the highest zero ratio and consistently outperforms other methods in unlearning performance. These results underscore the advantage of higher zero-out ratios achieved by NegMerge, enabling superior unlearning effectiveness compared to alternative methods.
>
> | **Method**       | **Zero Ratio** | **Df↓**   |
> |-------------------|----------------|-----------|
> | Single best  | 47.55    | 23.63 |
> | MagMax       | 47.55    | 25.24 |
> | TIES         | 51.27    | 26.21 |
> | NegMerge     | 92.94    | 20.76 |
>
> We have included this discussion in Appendix F. This significantly enhances the quality of our paper. Thank you for the constructive feedback.

---

> ### Author Response · Authors · 2024-11-25
> **To Reviewer AEkn (4/6)**
>
> **Q3) Could you provide statistics on the proportion of the merged task vector that becomes negated? Especially, when tuning the number of task vectors (e.g. 5, 10, 20, 30), would that have a significant impact on the zero ratio of the merged vector?**
>
> **A3.** Thank you for your insightful question. To address this, we conducted experiments by varying the number of task vectors used for merging (e.g., 5, 10, 15, 20, 25, 30). The experiments were conducted three times per dataset, and the table reports the averages and standard deviations (std) from these runs. These results were obtained using the ViT-B-32 architecture and vanilla task arithmetic. The table presents the averaged results across all datasets, and the full results for each dataset are included in the paper Table C1 and C2).
>
> | # of Task Vectors | Zero Ratio (std) | Df↓ (std) | Dr↑ (std) |
> |-------------------|----------------------|----------|----------|
> | 30               | 92.94               | 20.76    | 60.36    |
> | 25               | 92.29 (0.06)        | 20.86 (0.12) | 60.38 (0.04) |
> | 20               | 91.47 (0.09)        | 20.72 (0.20) | 60.33 (0.04) |
> | 15               | 90.04 (0.17)        | 20.97 (0.23) | 60.35 (0.10) |
> | 10               | 87.37 (0.35)        | 21.21 (0.40) | 60.45 (0.10) |
> | 5                | 80.29 (0.32)        | 22.60 (0.43) | 60.73 (0.10) |
>
> The results indicate that as the number of task vectors increases, the proportion of masked weight elements (Zero Ratio) also increases. Simultaneously, we observed consistent improvements in unlearning performance (Df↓). This suggests that utilizing more task vectors enables more precise identification of the weight elements that need to be modified for effective unlearning. Ultimately, the task vectors generated through our method lead to changes in only 5–10% of the weight elements in the original model via negation. These findings strongly support our argument that not all weight elements are critical for unlearning, and a small subset is sufficient to achieve strong unlearning performance.
>
> Moreover, these experiments also provide evidence that our technique maintains the performance of the retain set effectively. By selectively targeting only the weight elements essential for unlearning, our method minimizes the impact on the retain set, resulting in better preservation of its performance compared to other merging techniques. For more details, please see Appendix C.
>
> Your constructive feedback has significantly improved our understanding of our method and its presentation. Once again, we deeply appreciate your thoughtful input.

---

> ### Author Response · Authors · 2024-11-25
> **To Reviewer AEkn (5/6)**
>
> **Q4) How do you decide on the hyperparameters? Is it sensitive to choose the hyperparameters to create a set of task vectors?**
>
> **A4.** Thank you for your thoughtful question. The primary objective of our study was to demonstrate the effectiveness of model merging in enhancing machine unlearning. To the best of our knowledge, no prior research has shown that model merging can improve unlearning performance. As such, we focused on validating this potential rather than identifying the optimal strategies for constructing effective model pools.
>
> Aligned with this goal, we adopted straightforward approaches depending on the type of experiment. For CLIP-based experiments, we adjusted data augmentation intensities, while for general image classification experiments, we fine-tuned basic hyperparameters such as weight decay, learning rate, and label smoothing. We believed that demonstrating the effectiveness of such simple methods could establish a baseline for the potential improvements in unlearning performance.
>
> While exploring more advanced strategies for constructing model pools is undoubtedly an important research direction, we intentionally left this as a subject for future work at the time of submission. That said, the reviewer’s question prompted us to consider the impact of different model pools on unlearning performance, which we realized could significantly enhance our paper.
>
> To address this, we conducted additional experiments:
> 1. *CLIP fine-tuning*: We varied hyperparameters (e.g., RandAugment, weight decay, learning rates, and label smoothing) and generated results using seven different model pools. ViT-B/32 is used.
> 2. *Standard image classification*: We adjusted RandAugment parameters to explore various model pools. ResNet18 and CIFAR-10 are used.
>
> ---
>
> **CLIP Experiments - Detailed Results Reported**
>
> | Method          | Cars (Df↓)  | DTD (Df↓)  | EuroSAT (Df↓) | GTSRB (Df↓)  | MNIST (Df↓)  | RESISC45 (Df↓) | SUN397 (Df↓)  | SVHN (Df↓) |
> |-----------------|-------------|------------|---------------|-------------|-------------|----------------|--------------|------------|
> | Single Best Model | 28.62      | 28.03      | 7.66          | 5.31        | 14.56       | 27.19          | 51.38        | 6.75       |
> | NegMerge        | 27.42      | 26.80      | 7.03          | 4.82        | 12.89       | 18.23          | 48.73        | 6.71       |
>
> - *Model pool configuration*: randaug (n:1–2, m:1–10), lr (1e-05, 5e-06, 1e-06), wd (0.01, 0.1) – 8 models.
>
> | Method          | Cars (Df↓)  | DTD (Df↓)  | SUN397 (Df↓) |
> |-----------------|-------------|------------|--------------|
> | Single Best Model | 33.52      | 29.14      | 51.36        |
> | NegMerge        | 30.33      | 26.43      | 47.94        |
>
> - *Model pool configuration*: lr (1e-04, 1e-05, 5e-05, 5e-06), wd (0.01, 0.1), ls (0, 0.1) – 16 models.
>
> **CLIP Experiments - Seven Different Model Pools (Avg Results Reported)**
>
> | Pool         | Pool 1 (Df↓) | Pool 2 (Df↓) | Pool 3 (Df↓) | Pool 4 (Df↓) | Pool 5 (Df↓) | Pool 6 (Df↓) | Pool 7 (Df↓) |
> |--------------|--------------|--------------|--------------|--------------|--------------|--------------|--------------|
> | Best         | 23.63        | 22.80        | 23.21        | 23.21        | 24.05        | 21.31        | 21.19        |
> | NegMerge     | 20.76        | 21.69        | 21.56        | 22.23        | 22.65        | 19.37        | 19.08        |
>
> - *Model Pool Configurations*:
>   - Pool 1: randaug (n:1–3, m:1–10) – 30 models.
>   - Pool 2: lr (1e-04, 1e-05, 5e-05, 5e-06), wd (0.01, 0.1), ls (0, 0.1) – 16 models.
>   - Pool 3: lr (1e-04, 1e-05, 5e-05, 5e-06), wd (0.01, 0.1) – 8 models.
>   - Pool 4: lr (1e-05, 5e-06, 5e-05), wd (0.01, 0.1) – 6 models.
>   - Pool 5: randaug (n:1–2, m:5,10), lr (1e-04, 1e-05, 5e-05, 5e-06), wd (0.01, 0.1), ls (0, 0.1) – 64 models.
>   - Pool 6: randaug (n:1–2, m:5,10), lr (1e-04, 1e-05, 5e-05, 5e-06), wd (0.01, 0.1) – 32 models.
>   - Pool 7: randaug (n:1–2, m:1,5,10), lr (1e-05, 5e-06, 1e-06), wd (0.01, 0.1) – 8 models.
>
> **Standard Classification Experiments**
>
> | Method                | Df | Dtest | MIA  | Avg. Gap |
> |-----------------------|--------|----------|------|----------|
> | Retrain               | 94.76  | 94.26    | 12.88| 75.48    |
> | Single Best Model     | 95.88  | 91.31    | 9.58 | 2.40     |
> | NegMerge     | 95.76  | 91.03    | 10.76| 2.14     |
>
> - *Model pool configuration*: randaug (n:1, m:1–5) – 5 models.
>
>
>
> ---
>
> As shown in the tables, unlearning performance consistently improves compared to the baseline methods. These results highlight that while the degree of improvement may vary depending on the model pool, using multiple models consistently provides more stable performance gains than relying on a single model.
>
> We have added the above results in Appendix D. We believe that exploring optimal model pool construction for more effective unlearning is a promising direction for future research. Thank you again for raising this insightful question.

---

> ### Author Response · Authors · 2024-11-25
> **To Reviewer AEkn (6/6)**
>
> **Q5) What're the recommendations for a practitioner to create variants?**
>
> **A5.** Our method relies on the knowledge encoded in each task vector. Based on our observations, poorly constructed task vectors—such as those trained with unreasonable weight decay—can result in unreliable knowledge and introduce noise into the process. To mitigate this, we recommend constructing a task vector pool using reasonable and functional hyperparameters, ensuring the vectors are reliable and contribute effectively to unlearning.
>
> Furthermore, our method inherently produces a model with a well-optimized retain loss. This aligns with one of our core assumptions: the merged model should preserve performance on the retain set. Practitioners can leverage this property by using the retain loss as a guiding signal during the model generation phase, enabling more effective model merging through better monitoring and optimization of retain set performance. While we have not yet explored this idea, we view it as an exciting direction for future research. We have added this discussion in Appendix D.2. Thank you for raising this question.

---

> > ### Comment · Reviewer_AEkn · 2024-11-28
> >
> > I thank the authors for their thorough response, which has addressed my concerns. I will maintain my original positive rating of 6, as the paper is above the acceptance threshold.

---

> > > ### Author Response · Authors · 2024-11-29
> > > **Sincere Appreciation for Your Valuable Feedback**
> > >
> > > Dear Reviewer AEkn,
> > >
> > > Thank you for your kind response and for maintaining a positive assessment of our paper. We sincerely appreciate the time and effort you have dedicated to reviewing our work, as well as the valuable insights you have shared, which have been crucial in enhancing the quality of our submission.
> > >
> > > Should you have any additional suggestions or concerns that you believe require our attention, we would be more than willing to address them. We are fully committed to upholding the highest standards in our research and submission.
> > > Once again, thank you for your thoughtful review and kind support.
> > >
> > > Best regards,
> > >
> > > NegMerge Authors

---

### Author Response · Authors · 2024-11-25
**General Response**

We thank all the **Reviewers AEkn, s5f2, EBwf, 6Rn3** for their thorough and constructive reviews with valuable and insightful advice. We appreciate the positive comments from all reviews:

**Reviewer AEkn** - 1) Importance of addressing machine unlearning. 2) Simple yet effective approach. 3) Use of MIA metrics to evaluate privacy protection in addition to classification performance.

**Reviewer s5f2** - 1) Computational efficiency of the merging step. 2) Robust numerical results supporting our claims. 3) Concrete evidence provided by our experiments.

**Reviewer EBwf** - 1) Simple yet effective and intuitive approach. 2) Practicality and versatility of our approach through experiments across various architectures, emphasizing its applicability.

**Reviewer 6Rn3** - 1) Superiority of using multiple models over a single model. 2) Simple yet effective approach.

We have made every effort to address all the comments by carefully revising the paper. Throughout the revised paper, we highlighted the newly added or edited materials in blue. Please refer to the following summarization of the materials as follows:

- Provided theoretical insights are provided in Appendix F (**Q2** by **Reviewer AEkn**, **Q1** by **Reviewer EBwf**).
- Added experiments and analysis on the ratio of zeroed elements in Appendix C (**Q3** by **Reviewer AEkn**).
- Added experiments that show the robustness of the method on a larger model pool in Appendix D.1 (**Q4** by **Reviewer AEkn**, **Q5** by **Reviewer 6Rn3**).
- Provided recommendations for creating variants in Appendix D.2 (**Q5** by **Reviewer AEkn**).
- Clarified that it is appropriate to compare the effectiveness of techniques based solely on their forget set performance for the CLIP unlearning scenario  in Section 4.2. (**Q2** by **Reviewer EBwf**).
- Added experiments on a wider range of architectures and datasets in Appendix A.3 (**Q3** by **Reviewer EBwf**).
- Clarified the minimal impact of freezing the final layer of CLIP on performance in Section 4.2 (**Q4** by **Reviewer EBwf**).
- Clarified the memory and computational efficiency of the method in Appendix E (**Q5** by **Reviewer EBwf**, **Q3** by **Reviewer 6Rn3**).
- Added an ablation study to derive the improved final task vector in Appendix A.2 (**Q4** by **Reviewer 6Rn3**).

Detailed responses to other comments are left in individual comments.

---

### Meta-Review · Area_Chair_QDUM · 2024-12-21

**Metareview:**

This paper addresses the machine unlearning problem, aiming to selectively remove knowledge from a pre-trained model without retraining. The paper proposes generating multiple task vectors using different hyperparameters based on a forget set and merging them efficiently. Experimental results highlight the importance of elements with consistent signs in the task vectors.

After the rebuttal, this paper received four scores slightly above the borderline, but there are still some important concerns that remain unaddressed.

The authors state that the motivation for effective unlearning is that "the fine-tuned model should maintain high performance on the forget set while avoiding degradation on the retain set." However, I believe this is not the core objective of unlearning. As discussed in prior work, such as Task Arithmetic, the key goal of unlearning is for the final model (after applying unlearning techniques) to maintain performance on the retain set while effectively removing knowledge related to the forget set. Therefore, avoiding degradation on the retain set is crucial for the final model, not necessarily for the fine-tuned models used during intermediate steps. Section 4.3 does not adequately address this distinction, and other methods have demonstrated good results even when remaining accuracy is relatively low.

While the paper proposes an intriguing approach of aggregating only same-sign parameters for unlearning, several critical questions remain unanswered, leaving some uncertainty about the true impact of this method: Does obtaining fewer parameters with the same sign correlate with poorer unlearning? Does obtaining more same-sign parameters correlate with better unlearning? Won't parameters with opposing signs cancel out to some extent in expectation? If so, what degree of cancellation is significant for effective unlearning?
It appears that this paper essentially proposes averaging k models to obtain the unlearning task vector. However, the theoretical depth and contribution of this approach to unlearning remain open for further discussion.

Finally, regarding inference time complexity, the paper emphasizes that the task vector acquisition phase can reduce complexity to O(m) compared to Task Arithmetic (which involves m hyperparameters with n models). However, this paper requires a much larger n (e.g., 30 models), and the computational cost of fine-tuning 30 models far outweighs the evaluation on the forget set.

Therefore, the ACs believe this paper still requires further theoretical justification to fully address the remaining concerns.

**Additional Comments On Reviewer Discussion:**

Although all four reviewers thoroughly reviewed the authors' rebuttal and provided feedback with scores around the borderline, there are still some concerns that need to be addressed. Therefore, the ACs recommend rejecting the paper.

---

### Decision · Program_Chairs · 2025-01-22

Reject